# Wireless, battery-free, and real-time monitoring of water permeation across thin-film encapsulation

Massimo Mariello ⬡[1,2,5] ✉, James Daniel Rosenthal[1], Francesco Cecchetti[3], Mingxiang Gao[4], Anja K. Skrivervik[4], Yves Leterrier ⬡[2] & Stéphanie P. Lacour ⬡[1]

Long-term bioelectronic implants require stable, hermetic encapsulation. Water and ion ingress are challenging to quantify, especially in miniaturized microsystems and over time. We propose a wireless and battery-free flexible platform leveraging backscatter communication and magnesium (Mg)-based microsensors. Water permeation through the encapsulation induces corrosion of the Mg resistive sensor thereby shifting the oscillation frequency of the sensing circuit. Experimental in vitro and in-tissue characterization provides information on the operation of the platform and demonstrates the robustness and accuracy of this promising method, revealing its significance for in-situ real-time monitoring of implanted bioelectronics.

The reliability of soft and flexible implantable bioelectronics represents an urgent challenge for the development of novel functional devices with thin form factors and tissue adaptability. In order to increase the long-term functionality of chronic implants, they must be protected and insulated from the body environment, preventing any damages, short-circuits, and performance degradation. Instead of standard rigid metallic/ceramic packages[1], high-barrier flexible thin-film encapsulations (TFEs) are the most common strategy for this purpose, providing simultaneously hermeticity, flexibility, and compatibility with microfabrication[2–7]. TFEs are based on organic films (e.g., polyimide, parylene C, liquid-crystal polymers, etc.), inorganic films (e.g., aluminum oxide ($Al_2O_3$), silicon nitride ($SiN_x$)[8], silicon dioxide ($SiO_2$ or $SiO_x$)[9], silicon carbide (SiC)[8,9], or hybrid multilayer encapsulations made of atomic-layer-deposited (ALD) oxides and organic interlayers[2,4,10].

The main reason for barrier failure is the permeation of water molecules which can lead to short-circuits, corrosion, and delamination. Water permeation is usually quantified through the Water (Vapor) Transmission Rate (WVTR, WTR), defined as follows:

$$WTR = \frac{1}{S} \cdot \frac{dm_{H_2O}}{dt} \qquad (1)$$

where $m_{H_2O}$ is the mass of water permeated through the barrier and $S$ is the encapsulation area exposed to water. Encapsulation of bioelectronic implants calls for low values of WTR ($\leq 10^{-4}\ g/m^2/day$) and low thicknesses (<5 μm). However, in-situ quantification of the permeability of TFE is an unanswered challenge. Standard permeability-measuring technologies are based on bulky, expensive, and low-sensitivity permeation cells, which are not miniaturizable and do not allow direct measurement of WTR on real bioelectronic implants. The calcium (Ca) test[2,11–14] is not suitable because it requires very inert environments for Ca deposition; thus, it cannot be integrated into microfabrication processes and real implants. A novel method has been recently proposed[3], based on the use of corroding Magnesium (Mg) as a sensing element for water permeation. Mg is less sensitive to water than Ca, it can be deposited outside a glove-box environment and it is suitable for microfabrication processes and miniaturization. So far, there has been no evidence for the applicability of the Mg-based wireless method to flexible/stretchable substrates, that can provide information about the real evolution of water permeation in real implants[15].

In this article, we propose a novel, ultra-sensitive wireless platform using wireless, radio-frequency backscatter to quantitatively

---

[1]Laboratory for Soft Bioelectronic Interfaces (LSBI), Neuro-X Institute, École Polytechnique Fédérale de Lausanne, Geneva, Switzerland. [2]Laboratory for Processing of Advanced Composites (LPAC), École Polytechnique Fédérale de Lausanne (EPFL), Lausanne, Switzerland. [3]École Polytechnique Fédérale de Lausanne (EPFL), Lausanne, Switzerland. [4]Microwaves and Antennas Group (MAG), École Polytechnique Fédérale de Lausanne (EPFL), Lausanne, Switzerland. [5]Present address: Department of Engineering Science, Institute of Biomedical Engineering, University of Oxford, Oxford, UK. ✉e-mail: massimo.mariello@eng.ox.ac.uk; massimomariello@gmail.com

monitor the WTR of TFEs in situ and in real-time. It is based on active water-permeation sensing (WPS) and the corrosion of Mg thin films. The platform is wireless, operating off externally generated wireless power to transmit a frequency-modulated (FM) signal that is robust to changes in the electromagnetic environment. The key principle is the monitoring of the frequency shift of this modulated signal, as a result of a backscattering circuit tuned by the disposable corroding-Mg sensors. Therefore, it is possible to extract directly the value of WTR of the coating encapsulating the Mg sensor, knowing only the oscillation frequency as an operational parameter. Additionally, a specific design for a flexible dipole antenna allows wireless data communication, so that the platform can be miniaturized to a few cm², integrated with flexible substrates in polyimide[16], guarantee stability in physiological environments, and be fully employed for in vivo tests (subdermally or intramuscularly), as demonstrated by our implantation experiments in deeply anesthetized mice. The design is promising for tissue conformality and robustness, enabling a stable battery-free operation and represents a promising approach for the predictive evaluation of barrier encapsulations of soft bioelectronics.

## Results

### Tunable backscatter circuit: design and working principle

The proposed design leverages backscatter communication to wirelessly provide a quantitative measure of the WTR for TFEs. The design achieves robust (stably operating) wireless and battery-free operation for barrier assessment of encapsulations in both static and dynamic testing conditions, i.e., in vitro and in vivo. Wireless backscatter communication is used to reduce the power consumption of the device while enabling fully wireless operation. Unlike a conventional radio that modulates a locally generated RF oscillation, backscatter communication transmits data by selectively reflecting the externally generated carrier wave[17,18]. In a backscatter communication system, there are two main components, called the interrogator (or reader) and the tag, as illustrated in Fig. 1. To transmit data, the tag only needs to modify the impedance presented to the antenna, thus modulating the complex-valued power wave reflection coefficient:

$$\rho_i = \frac{Z_i - Z_a^*}{Z_i + Z_a} \tag{2}$$

where $Z_i$ represents the $i$th impedance presented to the antenna, $Z_a$ represents the input impedance of the antenna, and $(\cdot)^*$ is the conjugate operator[19]. In this way, backscatter communication reduces the power consumption of the tag by removing the power-hungry components of an RF transmitter: the RF frequency synthesizer and the RF amplifier.

In the proposed device, we implemented a pseudo-analog backscatter modulation that converts the resistance of Mg test sensors to a

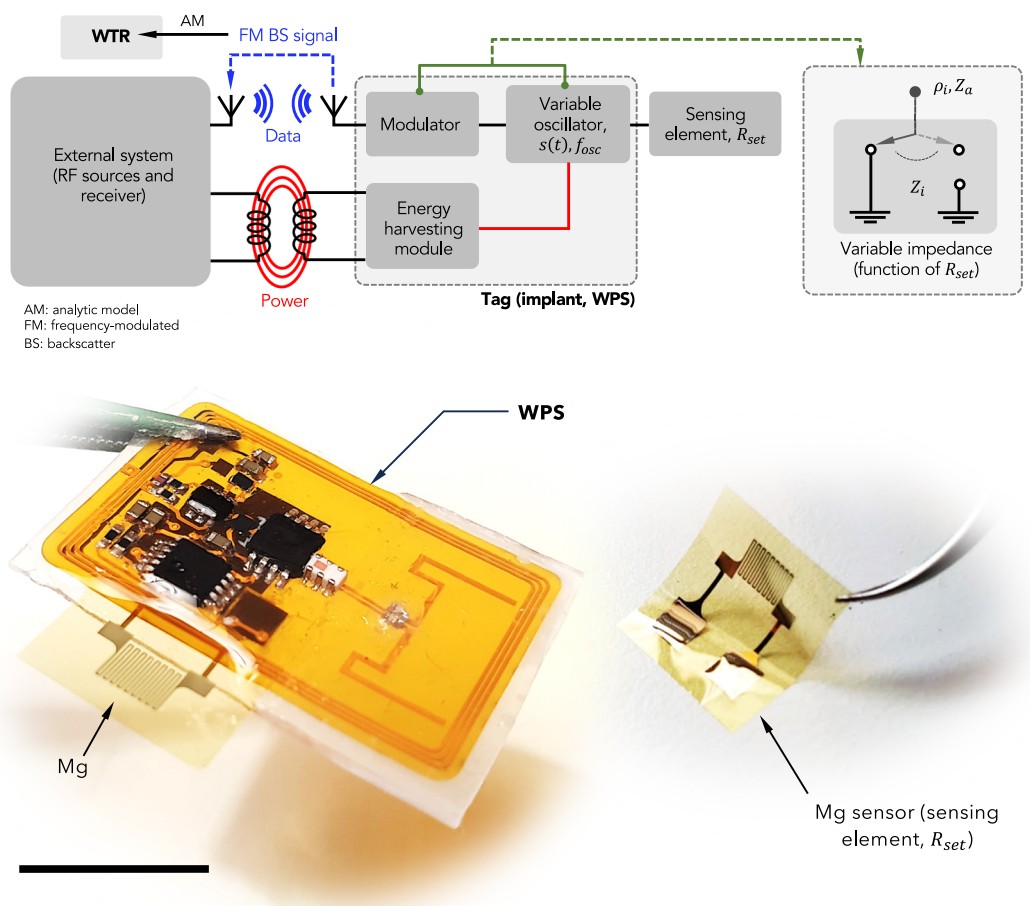

**Fig. 1 | Illustration of the architecture of the proposed backscatter communication system between a reader (in the external system) and a tag (implant, WPS).** The incident unmodulated signal coming from the RF source in the external system is used through the wireless powering unit to supply power to a variable oscillator, controlled by the variable impedance of a sensing element. The oscillator allows the modulator to provide a backscattered modulated signal transmitted wirelessly to the reader. The photo depicts the developed wireless platform, together with the Mg sensing element used to modulate the backscatter signal. Scale bar: 1 cm.

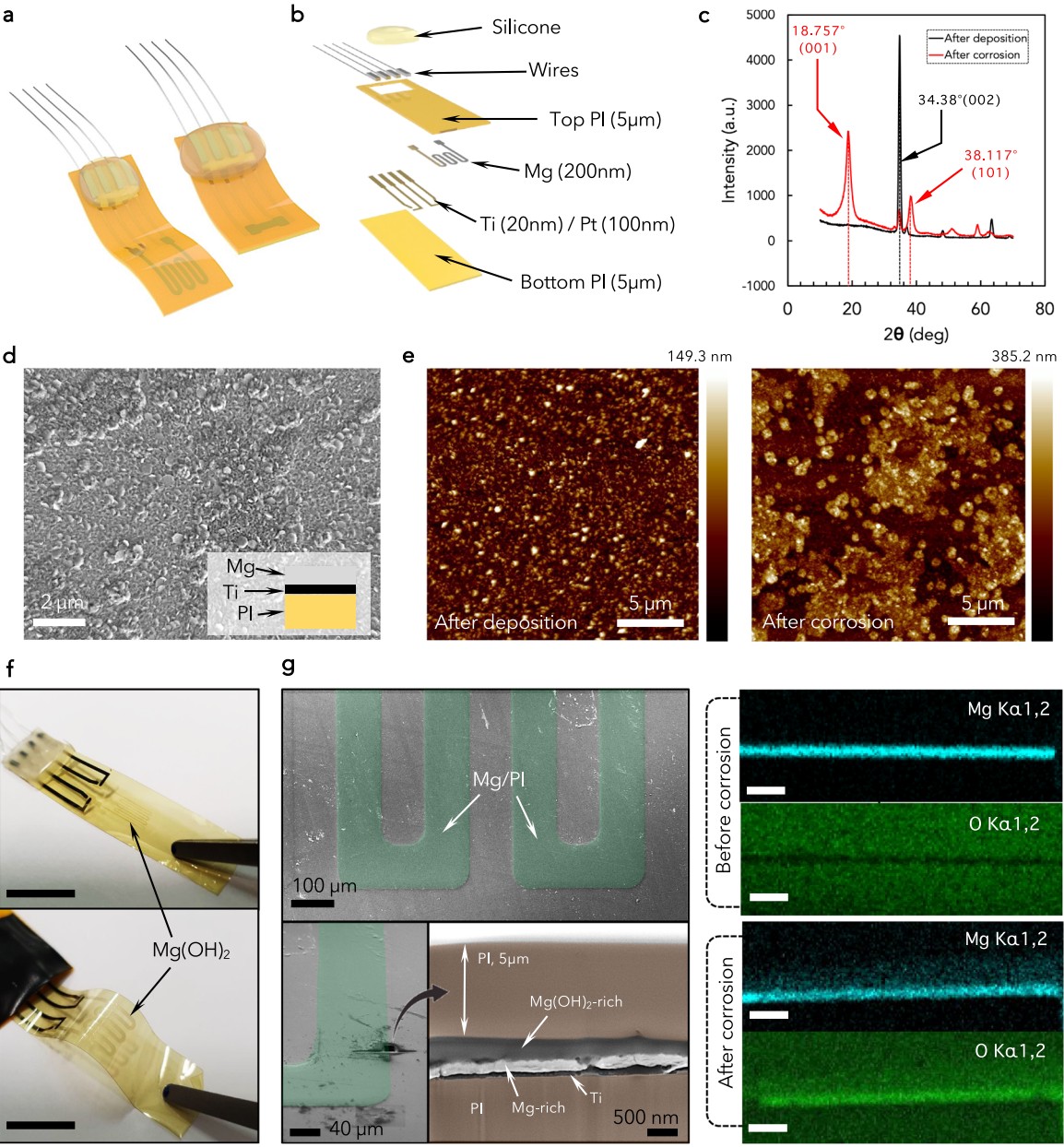

**Fig. 2 | Characterization of Mg flexible test sensors. a, b** Schematics of the test sensors. **c** XRD spectrum of the Mg thin film after the thermal evaporation and after the completion of the corrosion process. **d, e** SEM and AFM images of the Mg surface morphology. **f** Resulting devices after the complete Mg corrosion process. Scale bar: 1 cm. **g** FIB milled cross-section of an Mg test sensor, showing the appearance of Mg hydroxide due to the corrosion process, confirmed by the EDX analyses. Scale bar for EDX images: 1 μm.

FM backscatter signal, similar to the design in ref. 20. The resistance of Mg films can be tracked in real-time as they corrode by measuring the instantaneous frequency of the backscattered signal using an RF spectrum analyzer. The device is based on a square-wave oscillator whose output is:

$$s(t) = \Pi\left(2\pi f_{osc}t\right) \qquad (3)$$

where $\Pi(\cdot)$ is a square-wave signal, and $f_{osc}$ is the fundamental frequency of the square wave that is tuned by the resistance of the Mg test sample, $R_{set}$ as described ahead in the next section. The oscillator square-wave output controls a single-pole-dual-throw CMOS RF Switch (Analog Devices ADG919) that toggles the impedance presented to the device's antenna at $f_{osc}$ between two distinct values (Fig. S2). When the external carrier wave impinges upon the device's antenna, the wave is backscattered with a FM backscatter signal that appears as a tone. As $R_{set}$ varies during corrosion, the oscillator's frequency, $f_{osc}$, decreases and from this, the WTR value can be extracted through an analytical model.

**Microfabricated Mg test sensors**

Figure 2a, b shows an exploded view of the flexible Mg test sensors, with an indication of all the functional layers. The fabrication process was optimized to preserve the integrity of Mg after its deposition: it consists of sequential steps of thin-film deposition by DC reactive sputtering, thermal evaporation, and UV photolithography patterning followed by wet and dry etching (Fig. S3a–p, Supplementary Material). ~200-nm thick Mg films were deposited onto the cured flexible 5-μm

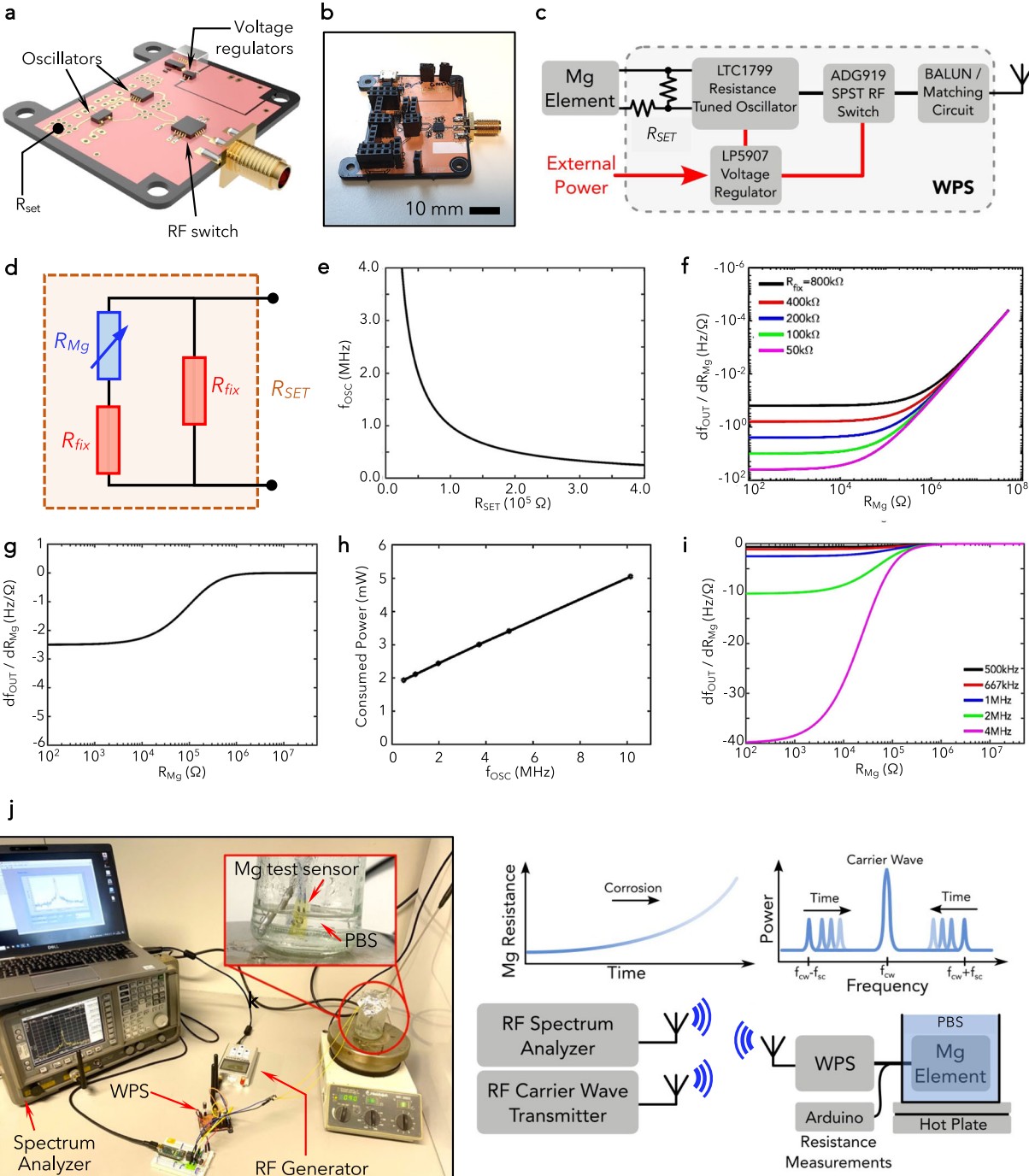

**Fig. 3 | Wireless sensing platform. a, b** Illustration and photograph of the rigid WPS system with an indication of the set resistance, the RF switch, the oscillator, and the supply voltage regulators. **c** Schematic of the WPS. **d** Parallel configuration for the set resistance with indication of $R_{fix}, R_{Mg}$. **e–i** Constitutive curves for the oscillator used in the WPS. **j** Experimental setup used for the calibration of the WPS and the WTR measurements with the Mg test sensors. On the right, the scheme with the main connections of the constitutive elements (RF carrier wave transmitter, RF spectrum analyzer, WPS, Mg element) is illustrated. The inset plots illustrate qualitatively the evolution of the Mg element's resistance and the corresponding change in the frequency position of the sub-carrier peaks with respect to the carrier wave peak.

---

thick PI substrate with a 20 nm Ti adhesion layer. They exhibited a hexagonal crystallographic microstructure, confirmed by the XRD spectrum in Fig. 2c, and a crack-free morphology (see the Scanning electron microscopy (SEM) images in Fig. 2d). We measured sheet resistance of $(23 \pm 2) \times 10^{-2} \, \Omega \, \text{sq}^{-1}$, an electrical resistivity of $(4.6 \pm 0.4) \times 10^{-8} \, \Omega \text{m}$ and a surface roughness Rms(sq) of $(18.4 \pm 5.2)$ nm (see the atomic force microscopy (AFM) topography images reported in Fig. 2e). The presence of high asperities, pinhole

defects or irregular morphologies in the Mg surface would lead to non-uniformities during corrosion, accelerating it in correspondence to these defects. Figure S4a shows the devices before peeling from the carrier wafer and after the fabrication. In order to evaluate the influence of the shape of the Mg patterns, we selected five different designs for the Mg test sensors (Fig. S3q and Table S1: they differ only for geometrical dimensions and shapes (stripes or serpentines), while the Mg thickness was kept constant at ~200 nm. The residual internal

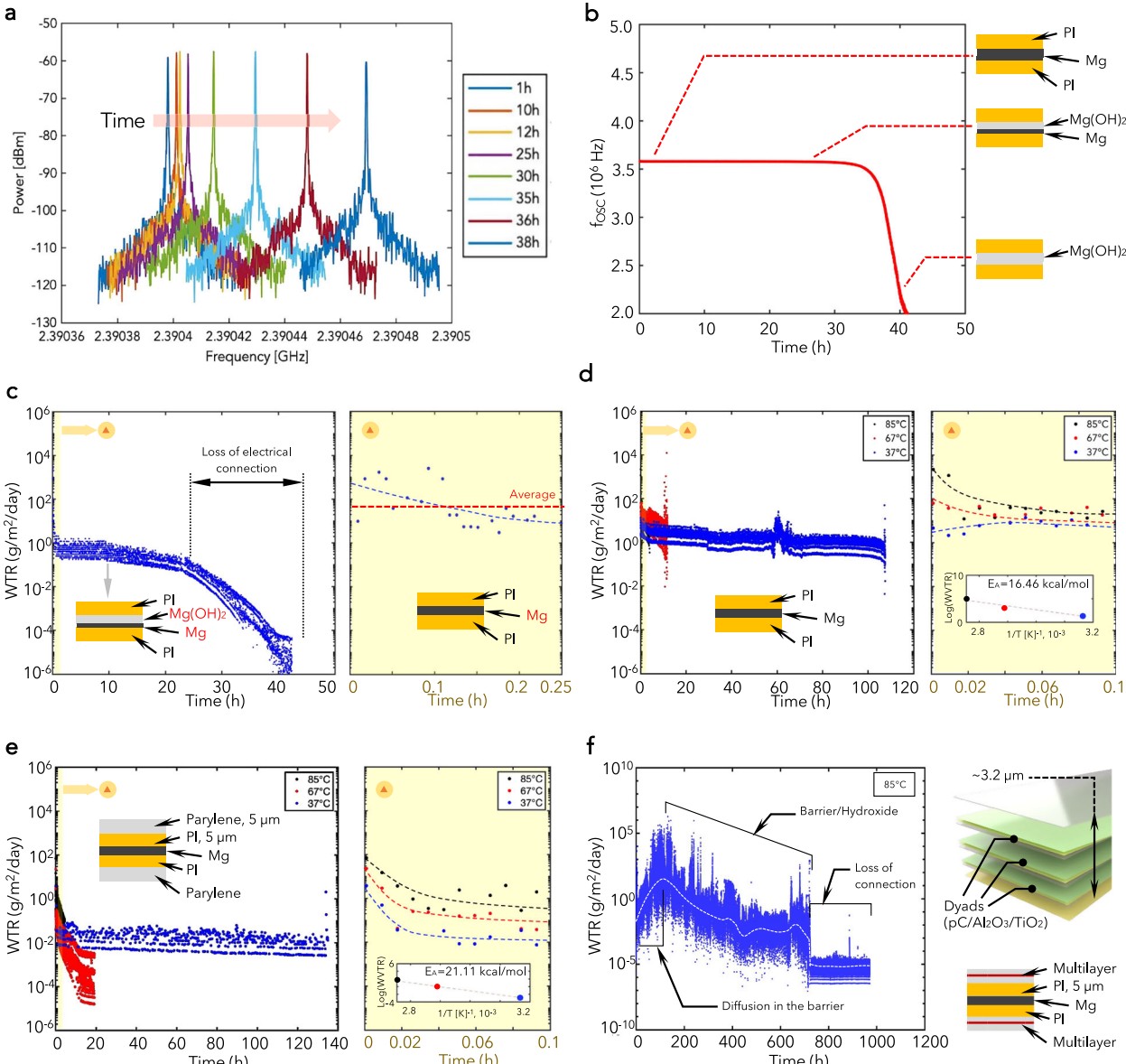

**Fig. 4 | Characterization with different barriers. a** Shift in the backscattered signal detected with a Mg test sensor. **b** Temporal evolution of the oscillation frequency changes in time due to the change in Mg resistance. **c** WTR curve of the PI encapsulation ($n = 4$). **d** WVTR curves for PI-encapsulated Mg test sensors at three different temperatures (37 °C, 67 °C, and 85 °C, respectively). The inset shows the corresponding Arrhenius plot. **e** WVTR curves for the Parylene/PI-encapsulated Mg test sensors at different temperatures. The inset shows the corresponding Arrhenius plot. **f** WVTR curve for the PI-based Mg test sensors encapsulated in a 3-dyad hybrid multilayer barrier, soaked in PBS at 85 °C.

stresses induced after the Mg thermal evaporation and the PI curing are released during the peeling from the Si carrier wafer. The overall rigidity of the PI/Ti/Mg/PI structure, evaluated by mechanical tensile tests (Fig. S4b), yielded Young's modulus of $(2.9 \pm 0.6)$ GPa, similar to that of PI (2.5 GPa) without any deposited layers, which confirms the extreme flexibility of the fabricated test sensors (see details in the Supplementary Material).

Following the corrosion process, the Mg patterns are converted into magnesium hydroxide ($Mg(OH)_2$), i.e., $Mg + 2H_2O \rightarrow Mg(OH)_2 + H_2$. The hydroxide appeared as transparent, as shown in Fig. 2f, where the Mg test sensors were removed from the testing solution (PBS), after the Mg corrosion. This process occurred through a progressive layering of a Mg-rich portion and a $Mg(OH)_2$-rich portion inside the sensitive Mg pattern: the first one was located at the bottom side, adjacent to the Ti seed layer and the bottom PI, whereas the second formed gradually on the top side, as observed in the focused-ion-beam

(FIB)-milled cross-section of the corroded Mg test sensors in Fig. 2g. The corresponding EDX colored map of the Mg and O elements shows that before corrosion the Mg layer appeared very bright while no oxygen was detected in its place. On the other hand, after corrosion, the brightness of the Mg layer became less intense and the O concentration increased. The process was also accompanied by an increase in the surface roughness of the Mg pattern, as observed in the AFM image of Fig. 2e: the oxidation appears non-uniform in the plane with the formation of islands and small spots which contributed to an increased Rms(sq) of $(50.5 \pm 9.9)$ nm.

## Assembly and calibration of WPS for measuring water permeation

The sensing circuit includes a tunable oscillator, components for supply voltage regulation (low-dropout (LDO) voltage regulator), an RF switch, and an SMA connector for the antenna. Figure 3a–c

illustrates the components of the rigid system. The switching element (RF switch) serves to modulate the input impedance and the reflection coefficient of the backscatter antenna, thus modulating the incoming RF carrier wave when reflecting it back. We calibrated the system using an LTC1799 tuneable oscillator (analog devices). The oscillation frequency $f_{OSC}$ is tuned with the set resistor $R_{set}$ connected between the power supply and the SET pin of the component (see Fig. 3d), according to the following expression (reported also in the Theory section of the Supplementary Material):

$$f_{OSC} = \frac{\psi}{R_{set}} \qquad (4)$$

where $\psi = 10\,k\Omega \cdot 10\,MHz$. $R_{set}$ is given by:

$$R_{set} = \frac{R_{fix}\left(R_{fix} + R_{Mg}\right)}{2R_{fix} + R_{Mg}} \qquad (5)$$

We opted for the parallel configuration for $R_{set}$ in order to confine the resistance magnitude between two limiting values during the Mg corrosion process. Choosing values of $R_{fix}$ in the range provided by the oscillators' manufacturers (3 kΩ–3 MΩ), when the Mg resistor is not corroded and its resistance is on the order of tens-of-ohms, the resulting $R_{set}$ value is around $R_{fix}/2$. On the other hand, when the Mg is corroded and its resistance is in the order of $10^6$ ohms, $R_{set}$ approaches the value of $R_{fix}$ (~55 kΩ). These two resistance limits translate to a well-defined range of oscillator frequencies for the backscattered peak. From this range of frequencies, the sensitivity, $S_{OSC}$, of the tunable oscillator with respect to resistance can be defined as a measure of how much the oscillator frequency changes based on variations in Mg resistance:

$$S_{OSC} = \frac{df_{OSC}}{dR_{Mg}} = \frac{df_{OSC}}{dR_{set}} \cdot \frac{dR_{set}}{dR_{Mg}} \qquad (6)$$

which can be combined with the Eq. 5 to yield:

$$S_{OSC} = -\psi \frac{R_{fix}^2}{R_{set}^2 \left(2R_{fix} + R_{Mg}\right)^2} \qquad (7)$$

When calibrating the system, we targeted a high $S_{OSC}$ to be able to detect small Mg resistance variations. Additionally, it is noteworthy that the value of the fixed resistor $R_{fix}$, besides defining the value of the oscillation frequency range, plays a role in affecting the sensitivity $S_{OSC}$: in particular, the latter increases with decreasing values of $R_{fix}$. We opted to perform the measurements with an oscillation frequency within the 3.5–4 MHz range, to achieve sufficiently high sensitivity $S_{OSC}$: as shown in Fig. 3e–i, correspondingly to such frequency, a shift in $f_{OSC}$ can be achieved of ~35 Hz for a resistance change of 1 Ω.

To calibrate the method and determine experimentally the WTR of some selected barrier encapsulations, we integrated the rigid WPS into an automated measurement system that enables the real-time recording of the evolving backscattered peaks over time for different testing conditions. The setup is shown in Fig. 3j with an indication of all the constitutive elements. We used the Mg test sensors as vehicles for the calibration and testing of the built setup. We soaked the devices in a PBS solution mimicking body biofluids and we set and controlled its temperature through a feedback-controlled hot-plate. We used an RF generator to send the carrier frequency wave, selecting $f_C$ at 2.394 GHz, to avoid interference within the industrial, scientific, and medical (ISM) band. We connected a spectrum analyzer (Agilent E4402B) to a laptop via a GPIB-USB-HS cable (National Instruments) to

detect and record the frequency spectra and to one monopole antenna: other two antennas were connected, respectively to the WPS and to the RF generator. For calibration purposes, we added to the system a microcontroller (Arduino) that enables the Mg test sensor to communicate either with the WPS or a voltage divider (100 Ω resistor in series with Mg, connected to a 3 V power supply): in this way, it was possible to perform simultaneously the frequency measurements and the monitoring of Mg resistance. Noteworthy, the latter was taken as control but the final aim of the proposed method is to rely only on wireless frequency monitoring. We set a Graphical User Interface (GUI) programmed in LabVIEW to control the continuous automated measurements (see the Supplementary Material). A 2.7 V power supply was used to power the rigid WPS and the voltage was stabilized with the LDO voltage regulator. The system recorded the spectrum centered at the backscattered peak with a sampling rate of 0.033 Hz (one recording every 30 s) and with a 100 kHz span, acquiring the position of the peak and the value of Mg resistance. The resolution bandwidth of the spectrum analyzer was set to 100 Hz, in order to have high-resolution measurements still keeping the duration of each recording below 5 s.

Connecting the rigid WPS to a digital multimeter, we detected a power consumption of 5 mW under a 2.7 V power supply. With the system described above, we determined the WTR of some selected encapsulations deposited on the Mg test sensors as explained next.

### Modeling and validation with selected barriers

Figure 4a shows the shift in the backscattered peak detected with a corroding Mg test sensor soaked in PBS at 85 °C and encapsulated in 5-μm thick PI: from this spectrum, it is possible to extract the evolution of the oscillation frequency in time and then the WTR of the barrier encapsulating the Mg sensor, according to the following expression (Supplementary Material):

$$WTR = -\frac{K \cdot \psi}{\left(2\psi - R_{fix}f_{OSC}\right)^2} \cdot \frac{df_{OSC}}{dt} \qquad (8)$$

where $K$ is a constant depending only on physical and geometrical parameters.

Figure 4b shows the oscillation frequency $f_{OSC}$ vs time for a corroding Mg test sensor: it can be observed that $f_{OSC}$ decreases in the time approaching the target value correspond to an infinite Mg resistance (i.e., $R_{set} = R_{fix}/2$). The resistance-vs-time plots in Fig. S5a, b follow the same trend, characterized by an initial linear increase, followed by a quick and abrupt variation[3]. Slight variations in the signal amplitude or in the oscillation frequency might be ascribed to the bending of the Mg sensors (see Supplementary Note 5). We can identify three sections in the WTR curve (Fig. 4c): in the first one, i.e., the first 15 min of corrosion reported in the zoomed plot on the right, the water diffusion through the barrier is the only occurring mechanism, the WTR takes on its maximum (integral-mean) value $(8.12 \times 10^1\,gm^{-2}\,day^{-1})$ with a root-mean-square deviation of $2.35 \times 10^1\,gm^{-2}\,day^{-1}$ (excluding 15% outliers). The results provided a standard deviation of $4.56 \times 10^1\,gm^{-2}\,day^{-1}$ [2]. The time interval chosen for the first range takes into account (with a certain overestimation) the amount of time necessary for water molecules to travel through the barrier and trigger the corrosion, as can be shown using the expression $t = h_B^2/(6D_B)$, where $h_B, D_B$ are the thickness and diffusivity of the barrier, respectively[21]. Then, the WTR value decreases due to the gradual formation of Mg(OH)$_2$, which provides an additional barrier to water permeation, reducing the water transmission rate. In the third section, the WTR stabilizes at $1.2 \times 10^0\,gm^{-2}\,day^{-1}$ which corresponds to the WTR order of magnitude of the Mg(OH)$_2$ only[3]. The final decrease of the WTR indicates the completion of the corrosion process.

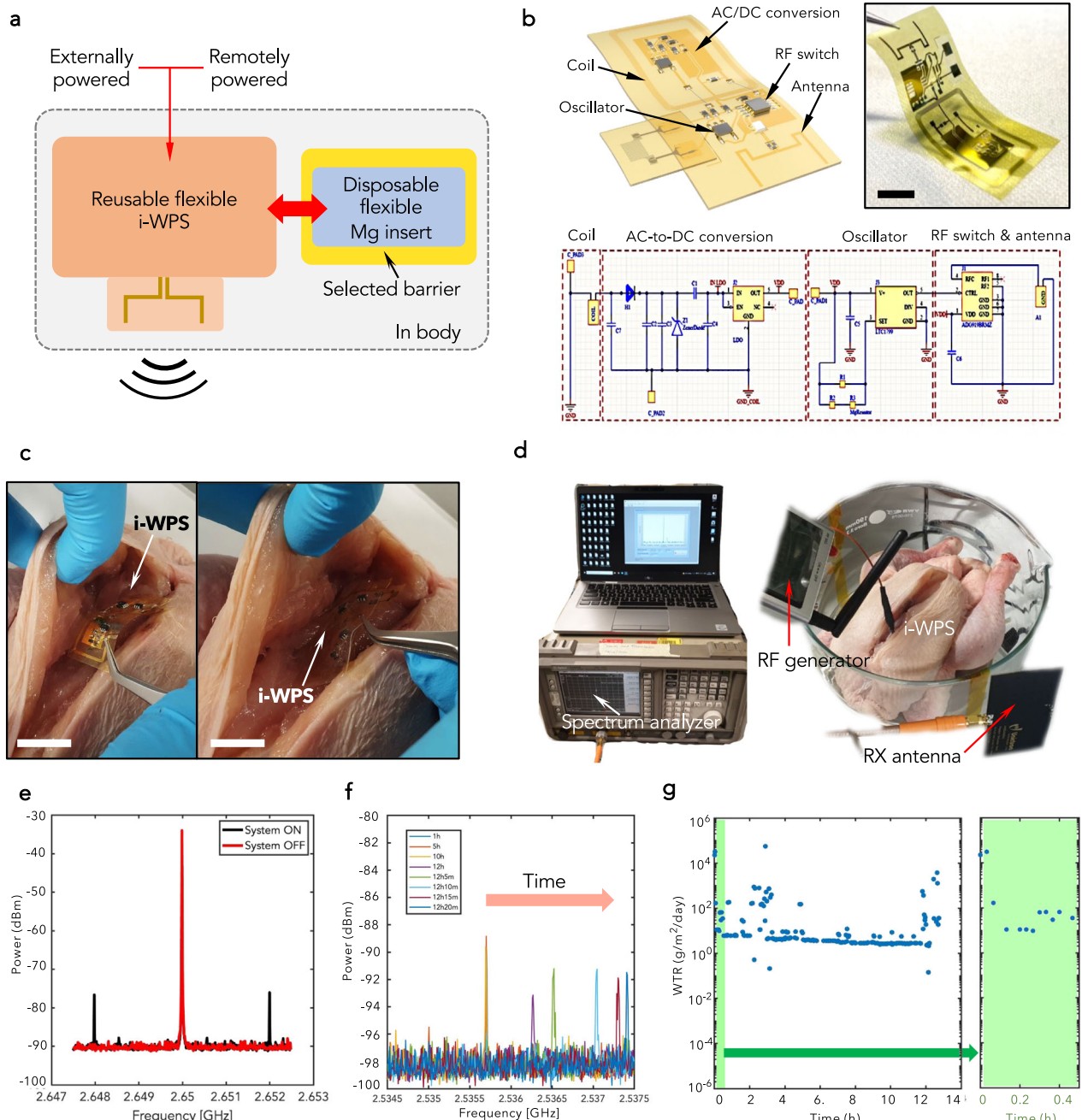

**Fig. 5 | Ex vivo deployment of the i-WPS. a** Conceptual scheme of the reusable i-WPS with a disposable Mg flexible insert. **b** Illustration and circuit of the wireless-remotely powered i-WPS. Scale bar: 5 mm. **c** Implantation procedure. Scale bar: 3 cm. **d** Experimental setup for the ex vivo demonstration. **e** Detected signal with the system on and off. **f** Evolution of the detected backscattered signal in time. **g** WTR curve for the PI-encapsulated Mg test sensor.

We determined the WTR for the five Mg designs (Fig. S7): under the same testing conditions and with the same encapsulation thickness, the design shape of the test sensor does not affect the value of the WTR (defined as a normalized quantity, according to the)), which resides in the same range of values (within the data scattering) for all of them in the initial range of measurement. However, the shape and in particular the width of the design influences the duration of the corrosion process (i.e., the critical time $t_c^B$: see the Supplementary Material for complete theoretical modeling of water-reactive diffusion with Mg corrosion): a wider pattern contains a higher amount of Mg, thus it completes its corrosion in longer times (Fig. S7).

Therefore, selecting one design (the narrow stripe or design 4 in Table S1), we performed the WTR measurements at three different temperatures: 85 °C, 73 °C, and 43 °C, respectively, and we estimated the activation energy of water diffusion into the encapsulation, according to the Arrhenius behavior (see the Supplementary Material). Figure 4d shows the WTR curves for the three selected temperatures: the higher the temperature, the higher the WTR because the diffusion (and correspondingly the corrosion process) is accelerated.

From the Arrhenius plot in the inset of Fig. 4d, we deduced an activation energy of 16.46 kcal mol$^{-1}$. This value is higher than what has been reported for polymer films[22–24], and this can be ascribed to the presence of the inorganic hydroxide layer, which starts forming in the initial range of measurement. In addition, soaking tests can favor the permeation of type II water into the polymer, yielding a higher activation energy if compared to tests in wet air[25]. We carried out the same

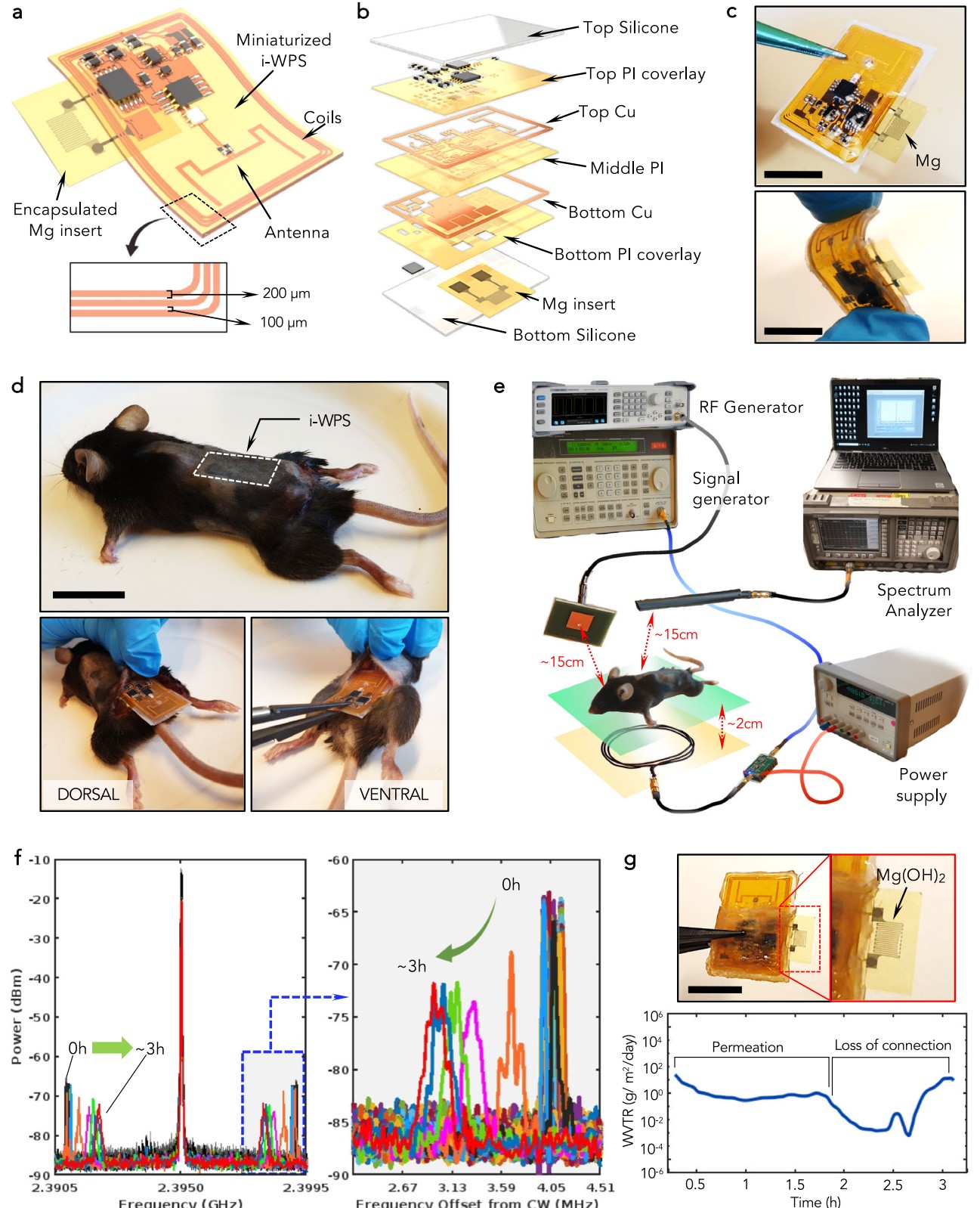

**Fig. 6 | Miniaturized i-WPS for in vivo-like demonstration. a–c** Illustration (**a**), exploded view (**b**), and photo (**c**) of the miniaturized i-WPS based on a multilayer compact architecture. Scale bar: 1 cm. **d** Photos of the implantation in a mouse body in dorsal/ventral positions. Scale bar: 2 cm. **e** Illustration of the setup used for the in vivo-like demonstration with an indication of the relative distances between the components. **f** Spectra of the detected signal evolving in time. **g** Frequency-time and WTR-time plots for the tested PI-encapsulated mini-i-WPS. The inset shows the mini-i-WPS after removal from the mouse body and after corrosion of the Mg insert (scale bar: 1 cm).

analysis with a second encapsulation (5-μm thick parylene C deposited by room-temperature chemical vapor deposition, RT-CVD), coating conformally the 5-μm thick PI. Figure 4e shows the results of the WTR measurements for this bilayer organic encapsulation: the value of the WTR at 85 °C is $9.2 \times 10^{0}\,\mathrm{gm^{-2}\,day^{-1}}$, lower than in the case of the only-PI encapsulated test sensor ($8.12 \times 10^{1}\,\mathrm{gm^{-2}\,day^{-1}}$). This is because the additional parylene layer introduces another barrier to water permeation. From the theory of water diffusion in multilayer laminates (see the Supplementary Material), we could extrapolate the transmission rate of parylene C only, i.e., $1.01 \times 10^{1}\,\mathrm{gm^{-2}\,day^{-1}}$, which is in accordance with the value presented in literature[2,26]. We derived also an activation energy for PI/Parylene encapsulation of 21.11 kcal mol⁻¹ (see the inset in Fig. 4e). The slightly higher activation energy indicates a higher barrier performance of the bilayer organic encapsulation compared to the PI.

A third example of encapsulation tested with the proposed WPS consisted of a hybrid organic-inorganic multilayer barrier. We characterized a multilayer architecture with repeated dyads made of three materials: 0.6-μm thick RT-CVD parylene C (organics), 30-nm thick $Al_2O_3$, and 5-nm thick titanium oxide ($TiO_2$) (inorganics deposited by atomic layer deposition (ALD)). The deposition process was performed in a single reactor by an external provider (Comelec SA, La Chaux-de-Fonds, Switzerland) and is described in the Supplementary Material All the materials were grown on an initial 5-nm thick ALD $Al_2O_3$ seed layer, deposited directly onto the PI of the Mg test sensors, while the multilayer structure was completed by a capping layer of 1.2-μm thick Parylene C: the former ensures good adhesion to the substrate, the latter protection and mechanical integrity (Fig. S9a, b)[4]. The soaking test in PBS at the accelerating aging temperature of 85 °C yielded the WTR curve reported in Fig. 4f. The PI/3-dyad encapsulation exhibited an (average) WTR of $8.2 \times 10^{-2}\,\mathrm{gm^{-2}\,day^{-1}}$ at the very beginning of permeation and a peak (average) value of $5.0 \times 10^{1}\,\mathrm{gm^{-2}\,day^{-1}}$. After this period, the combined $Mg(OH)_2$/PI/3-dyad barrier stabilized its WTR to a post-saturation value of $\sim 4.5 \times 10^{-2}\,\mathrm{gm^{-2}\,day^{-1}}$. Finally, the abrupt drop is due to the loss of electrical connection. The low values of WTR for this encapsulation, compared to those of standard polymers, are due to a defect-decoupling effect provided by the hybrid organic-inorganic multilayer structure (Fig. S9c)[2,4].

It should be noted that the proposed method is sensitive enough to discriminate between homogeneous diffusion through polymer encapsulations, and inhomogeneous diffusion through defective organic-inorganic multilayer encapsulations. In the latter case, inorganic layers contain inherent defects such as pinholes and microcracks that provide a pathway for rapid water permeation and localized Mg corrosion. These surface defects, especially pinholes, may also be present in polymer encapsulations as a result of non-ideal deposition processes. These two diffusion regimes are highlighted in Supplementary Note 9 and in the "Theory" section.

## Flexible implantable WPS (i-WPS) and in-tissue demonstration

Next, we converted the rigid WPS into a flexible i-WPS, which was microfabricated according to the process flow described in Fig. S11 (Supplementary Material). We propose the idea of a platform connected with disposable Mg sensor inserts coated with the selected encapsulation and soldered temporarily on the board. This concept is illustrated in Fig. 5a. We fabricated the inserts according to the same process flow used for the off-board Mg test sensors and we chose two sample designs (see Figs. S12a, b and S13a (Supplementary Material)). We designed (Altium and KLayout software) and fabricated two versions of i-WPS: (i) an externally-powered i-WPS and (ii) a wireless remotely-powered i-WPS. The latter included an inductive coil for wireless powering at 13.56 MHz, designed as described in Supplementary Material The coil had an inductance of 2.39 μH at the frequency of 13.56 MHz and was inductively coupled with an external transmitter coil to achieve wireless powering. A full-wave diode

rectifier was used to convert the RF AC voltage to DC. Figure S13b and Fig. 5b show the circuit schematics of the two i-WPS versions with an indication of all the electrical components. We designed the dipole antenna to resonate around 2.4 GHz when implanted in the tissue at around 3 mm depth: Fig. S13c shows the dimensions of the antenna with the corresponding plot of its scattering coefficient ($S_{11}$) between the port impedance and the network's input impedance (see Supplementary Material for the definition of $S_{11}$). The antenna was tested separately in a cuboid chicken phantom as illustrated in Fig. S14a, b, with resulting impedances at 2.45 GHz of $(15 + j7.5)\,\Omega$ and $(45 - j10.5)\,\Omega$ for the original and matched configuration, respectively (see the corresponding Smith chart in Fig. S14c). The matching resulted in optimal, minimizing the reflected power from the antenna (lower voltage standing wave ratio, VSWR). Figure S13d illustrates the geometrical parameters for the coil used for wireless powering, with the corresponding electromagnetic simulations and impedance plots reported in Fig. S17a, c. We next placed the platform intramuscularly in chicken meat in order to demonstrate the applicability of the system in biological tissue (Fig. 5c, d). Such resonating frequency favors keeping the size of the antenna small and saving space for the implantation. We checked the operation of the i-WPS, measuring a power consumption of 3.5 mW with a supply voltage of 2.8 V. When the i-WPS was turned on and powered, the backscattered peaks at $f_C - f_{osc}$ and $f_C + f_{osc}$ could be clearly observed (Fig. 5e). The oscillation frequency was set by the fixed resistors at ~2 MHz with $R_{fix} = 49.9\,\mathrm{k\Omega}$) and the RF generator was tuned to send 15 dBm carrier frequency at 2.650 GHz (resonance frequency) (see Fig. 5e). Then, we soldered a disposable Mg insert encapsulated in 5-μm thick PI (on both sides) on the dedicated pads on the i-WBS, and we performed the ex vivo measurements, as described in the Supplementary Material. In the ex vivo environment, the antenna was found to resonate at 2.54 GHz. Figure 5f shows the resulting evolution of the backscattered peak over time due to the corrosion of the Mg insert, whereas Fig. 5g depicts the evolution of the WTR values. Minimal alterations in the signal amplitude may be caused by small changes to the measurement environment, e.g., slight changes in the antenna orientation or variations in the dielectric properties of biological tissues caused by temperature changes, which may induce fluctuations in the path loss of the considered wireless link. However, this does not have an impact on backscatter communication and highlights the benefit of encoding information in the frequency domain rather than in amplitude. In this case, we extracted a WTR of $\sim 9 \times 10^{0}\,\mathrm{gm^{-2}\,day^{-1}}$, and the Mg insert reached the completion of corrosion in ~12 h, which is the same lag time for corrosion for in vitro testing at 85 °C, suggesting that the ex vivo tissue represents a more aggressive environment than a PBS physiological solution, owing to a diverse composition of the real biofluids (e.g., proteins, lipids, exudates, and other ions and molecules).

We finally performed an in vivo-like demonstration for the i-WPS system, optimizing the design of the implantable board as illustrated in Fig. 6a. This new design allowed the miniaturization of the i-WPS (mini-i-WPS), making it suitable for implantation in small rodents (mice, rats). Figure 6b highlights the various functional layers of the mini-i-WPS (PCBway): it consists of a bottom copper (Cu) coverlay with the contacts for the Mg test sensor, and a top Cu coverlay with the rest of the backscatter circuit and the dipole antenna. The coil for remote powering is patterned on both coverlay through via holes. An intermediate PI layer separates the two Cu coverlays. The mini-i-WPS is then encapsulated in a silicone coating to protect the electronic components during implantation. The final device is shown in Fig. 6c, whereas the full design of the coverlay is reported in Fig. S12c (Supplementary Material).

We implanted the mini-i-WPS subcutaneously in the body of mouse cadavers immediately after the animal death and kept at body temperature (37 °C). This allowed us to test the devices with accuracy, using the same operative conditions and exploring separately the

influence of several parameters (e.g., orientation of the antennas, orientation of the animal body, relative distance between the wireless components), using a host environment that simulated the living conditions. Generally speaking, death initiates a cascade of biochemical processes that affect the biophysical properties of most of the body tissues and internal organs. Thus, for specific implantable applications, such as optogenetics, electrical stimulation/recording, or ultrasound-based techniques, the implants' performances in ex vivo and in vivo conditions could differ[27]. However, the described method is reliable enough for measuring the predictive permeability of encapsulations subcutaneously (Fig. S15), with the purpose of establishing a quantitative correlation with in vitro accelerated aging tests. In fact, the resulting transmission rate derived from these tests can be considered reliably as the upper-limiting value for the barrier performances. More specifically, the implantation procedure is shown in Fig. 6d, while Fig. 6e depicts the setup used for the test. Using a PI-encapsulated Mg insert soldered to the silicone-encapsulated mini-i-WPS, the backscattered signal exhibited the temporal evolution shown in Fig. 6f, whereas Fig. 6g reports the plots of the sub-carrier peak position and WTR in time. It can be deduced that the in vivo-like performance of the PI encapsulation provided a WTR of $\sim 2 \times 10^0 \, \mathrm{gm^{-2} \, day^{-1}}$, similar to the tests performed in a chicken phantom. After the complete corrosion (a few hours long), the Mg insert appeared transparent as reported previously (see inset in Fig. 6g). Additionally, the test allowed us to assess the usability of the encapsulation for a small, compact mm-sized form factor, under a tissue-adaptive deformation. No significant differences in the detected signal intensities were observed by (i) changing the orientation of the mouse body and keeping the same relative positions between the wireless components (see Supplementary Movie), (ii) implanting the mini-i-WPS in different positions in the animal body, i.e., dorsal or ventral (see Fig. S19), or (iii) by changing the natural bending curvature of the animal body. In Supplementary Note 18, the contribution of bending, rotation, or tilting caused by any animal movement during testing is discussed.

Moreover, the in vivo-like demonstration was conducted for subcutaneous implantation, but deeply implanted bioelectronics may experience significant attenuation of wireless link efficiency due to lossy biological tissues, particularly in-body path loss. According to previous theoretical studies on implantable antennas[28,29], especially for deep-body implants, the in-body path loss can be analytically decomposed into three main contributions: (i) reactive near-field losses, (ii) propagating field loss, and (iii) reflection loss occurring at the tissue interface. The first two of them, in particular, will cause more losses as the implantation depth increases. As a result, the gain of the antenna decreases significantly from subcutaneous implantation to deep implantation, which poses a challenge to the link strength of wireless communications. However, this is only a change in signal strength and does not affect the oscillation frequency of backscatter communication. In addition, for the engineering design of implantable RF devices, it is often necessary to optimize the antenna structure and matching network according to the specific implantation location and packaging structure to reduce mismatch losses of the implantable antenna. Therefore, if the implant is designed for deep implantation, the impact of unnecessary reflection losses on wireless communications can be avoided by redesigning and retuning the antenna used in the implant, based on accurate 3D electromagnetic simulations including the host body model.

## Discussion

We proposed a novel wireless platform to assess the barrier lifetime in real time of TFEs for soft bioelectronic implants. The system is based on monitoring the frequency shift of the backscattering oscillator tuned by microfabricated Mg test sensors. Using an analytical model, it is then possible to extract directly the value of the WTR of the

encapsulation under test, depending only on an operational parameter, i.e., the frequency. The integration and miniaturization of all the components of this system into a single flexible platform allowed to design of a fully implantable (subdermally or intramuscularly) battery-free water-permeation device that enables the in situ dynamic interrogation of water ingress into any type of encapsulations.

Further optimization of the system regards the design of the remote powering unit, which should provide a high enough inductance, or the design of the AC/DC circuitry composed of the voltage rectifier and regulator, in order to assure a supply voltage as stable as possible for maximizing the signal-to-noise ratio. More sophisticated topologies for the rectifying circuit can be selected, such as a two-stage differential Greinacher configuration, widely employed in energy harvesting circuits for biomedical applications[30]. Alternatives for low-power tuneable oscillators can also be taken into consideration, as well as the most optimal configuration for the resistance network. The low thickness, flexibility, and conformality of the device favor the implantation in any type of tissue and/or animal models: further miniaturization can be possible for implantation in small rodent animals, such as rats, through shrinkage of the size of the inductive coil or employing a novel design for the implantable antenna. A convenient solution could be to opt for a single-frequency system, i.e., to use the same antenna to receive power and to backscatter the RF carrier. The dual frequency design is preferable when high data rates are required, but in the case of corrosion detection, the low amount of data would allow duty-cycling between power scavenging and data transmission at a single microwave frequency.

Additionally, the main advantage of the proposed WPS against other existing technologies is the continuous monitoring of the corrosion, encoded in the frequency shift of the backscattered peaks. This makes the method more robust against variability in the gain of the antenna owing to implantation or to a variable relative position between the reader and the tag, which happens in live-moving animals. Future perspectives and improvements can be envisioned in terms of miniaturization, integration into implants with optoelectronic functionalities, and full deployment in vivo, for which a comprehensive setup optimization will be required, including the design of a wireless freely-behaving-rodent cage to host the animals, a large transmitting coil wrapped around the sidewalls (see Supplementary Movies 2 and 3). Furthermore, the Mg corrosion is linked to the portion of TFE right above the Mg sensor, thus more sophisticated designs would include multiple Mg resistors spread all over the implant, in order to have a map of the encapsulation reliability. Additionally, the proposed system is active, with just a consumption of a few mW of power, and it allows a detailed, dynamic, robust, and reliable measure of the barrier's WTR.

In this context, the choice of using backscatter communication has been driven by its significant energy efficiency relative to conventional RF solutions, such as Bluetooth-low-energy (BLE), and its potential for far-field wireless communication as would be needed for in vivo experiments with animals in their home cages. BLE has been demonstrated in many small implants[31] and could be an option for a future embodiment of the device. For this work, as a proof-of-concept demonstration, analog backscattering of a subcarrier frequency provided an appealing balance between power consumption, measurement resolution, cost, and complexity. Near-field communication (NFC) could be considered as an alternative strategy for wireless operation and it also uses backscattering; however, it does so in the 13.56 MHz frequency band with near-field inductive coupling, requiring very close proximity for effective operation. As a result, using NFC communication in future applications with moving rodents could be problematic.

All the aforementioned features reveal a big potential for the use of this platform in bioelectronics research, especially for the predictive quality control of TFE adopted for biomedical implantable devices. Although this work is focused on the characterization of the WPS platform as a single implant, it can be easily integrated into more

complex wireless systems with other functionalities. The size of the design, in fact, has been driven by the need to use discrete components for prototyping. While we adopted specific frequencies for wireless power transfer and wireless data communication (i.e., 13.56 MHz and 2.4 GHz, respectively), the proposed concept is expected to work for higher frequencies, thus enabling further antenna miniaturization and high data rates (e.g., 5.8 GHz). Additionally, the components of the overall electronics are small and commercially available, lending themselves to easy miniaturization in an application-specific integrated circuit (ASIC). One possible embodiment would be to create an ASIC of the design as a small sticker that could be independently added onto an implant, such as an RFID tag integrated into clothing tags. Furthermore, the Mg resistor, which in the current design is shaped as an external insert with macroscopic dimensions, can be miniaturized and included in the overall circuitry on a microscopic scale (e.g., as a small serpentine-like resistor with ~1–10 μm trackwidth). In this way, a complete bioelectronic implant would include this compact module to monitor the health of the encapsulation without interfering with the rest of the functional system. Thanks to developments creating BLE-compatible backscatter systems[17,32,33], an optimized design could be envisioned that could be interrogated and read with smartphones. Overall, the WPS platform offers a practical tool for in-situ monitoring of the permeability of existing ultrathin TFEs and for new advanced ultra-high-barrier encapsulations, whose development and implementation are urgently needed for chronic applications.

## Methods

### Design and fabrication

**Fabrication of flexible Mg test sensors.** The fabrication process of the flexible test sensors is illustrated in Fig. S3a−p and starts with the spin-coating of a 5-μm thick layer of polyimide (HD Microsystems GmbH, catalog no. PI2611) on a 4-inch silicon wafer at 2500 rpm, followed by soft bake (70 °C for 3 min, 110 °C for 3 min) and curing for 2 h at 300 °C in a N2-filled oven. The interconnection tracks in Ti (20 nm)/ Pt (150 nm) were sputtered (Alliance Concept AC450) and patterned through liftoff with a first photolithography (MicroChemicals, catalog no. AZ1512). Mg thin films (200 nm) were thermally evaporated (Alliance Concept E300) and patterned in the shape of serpentines or stripes with different widths through a second photolithography-liftoff step. A second 5-μm thick PI layer was spin-coated on top of the Mg patterns and cured in the same way as the first one, then it was patterned through photolithography (MicroChemicals, catalog no. AZ10XT) and RIE-etching (Corial 210IL) to expose the contact pads. Wiring was carried out with sliver paste or low-temperature soldering onto the pads and finally, the connections were mechanically fixed with silicone° ped by laser cutting (Optec WS Turret 200) and peeled off from the Si carrier. The same process flow was adopted for the fabrication of the disposable Mg inserts used for the assembly of the flexible implantable system.

Five designs were selected for the patterning of Mg thin films and they are reported in Fig. S3q, whereas Table S1 summarizes the geometrical parameters of each design.

**Design of resistance-to-frequency conversion system.** The resistance-to-frequency converting wireless backscatter system is based around a variable frequency CMOS square-wave oscillator whose fundamental frequency is tuned by an external resistance. The square-wave oscillator output is used to actuate a single-pole-single-throw RF switch such that one of two discrete impedances will be presented to an antenna connected to the RF common port of the switch at the oscillating frequency.

The WPS system was designed in Altium (Altium LLC) schematic capture and circuit board layout software using commercial off-the-shelf parts, except for the implantable antennas (see Section S1.3).

We calibrated the WPS system considering two tunable oscillators, i.e., LTC1799 and LTC6902, (analog devices). The expression correlating the oscillation frequency $f_{OSC}$ with the set resistor $R_{set}$ can be written as follows:

$$f_{OSC} = \frac{\psi\alpha}{R_{set}} \qquad (9)$$

where $\alpha = 1,2$, respectively for LTC1799 and LTC6902.

$R_{fix}$ was chosen in the range provided by the oscillators' manufacturers (20 kΩ−400 kΩ for LCT6902 and 3 kΩ−3 MΩ for LTC1799). Using the oscillator LTC1799 allows to achieve a larger $S_{OSC}$ sensitivity as shown in Fig. S8, where the sensitivity curves for the two oscillators are reported for an oscillation frequency of 1 MHz. Concerning the power consumption, we compared the two oscillators in terms of dissipated power (see Fig. S8c): the power cost is similar, especially in the 4−6 MHz region, because LTC1799 requires half the value of $R_{fix}$ used for LTC6902, allowing a higher current flow (under the same supply voltage). Collectively, the previous considerations led us to select the LTC1799 oscillator for its smaller size and complexity along a higher sensitivity. We opted to perform the measurements with an oscillation frequency within the 3.5−4 MHz range, to achieve sufficiently high sensitivity $S_{OSC}$: as shown in Fig. S8b, correspondingly to such a frequency, a shift in $f_{OSC}$ of ~35 Hz for a resistance change of 1 Ω can be achieved.

The rigid and flexible printed circuit boards (PCBs) were supplied from PCBWay using their standard two-layer PCB processes. PCB assembly was performed using low-temperature lead-free solder paste and manually placing components by hand. Soldering was then performed using a hot plate.

**Design of implantable antennas.** To achieve backscatter communication from in-body implants, we designed flexible implantable antennas operating at the ISM band of 2.4−2.5 GHz.

Due to the high loss characteristics of the biological tissues surrounding the implants, the major limitation of the implanted antenna is their low radiation efficiency, resulting in tight link budgets[28,34,35]. However, due to the high water content of biological tissues, their high permittivity allows the antenna size to be effectively reduced, as the wavelength is shorter.

Considering the limited volume of the implant, especially the very thin thickness for subcutaneous implants, we adopted the design of a dipole antenna with meander lines (see Fig. 5e and Fig. S14b). In order to minimize the near-field losses of the implanted antenna[36] we set the total thickness of the encapsulation around the antenna to 1 mm. For the balanced feed port of the dipole antenna, we introduced a balun between the RF port and the antenna itself. Specifically, a multilayer chip balun (Johanson Technology, 2450BL15K100E) was used to convert the 50-Ω unbalanced RF port to the 100-Ω balanced coplanar stripline.

Another important point is the impedance matching circuit between the antenna and the feed line (i.e., the 100-Ω coplanar stripline). Unlike antennas operating in free space, the input impedance of implanted antennas is significantly affected by the surrounding biological tissues, especially when the dimension of the antenna encapsulation is electrically small. As shown in Fig. S14a, we measured the input impedance of the designed antenna implanted in a cuboid chicken phantom. According to the original impedance of the antenna (see Fig. S14c), a π-matching network was designed at the center frequency of 2.45 GHz, with the inductor of 4 nH and both capacitors of 1.2 pF (see Fig. 5e). The input impedance of the antenna after matching was measured using a vector network analyzer (Hewlett Packard, 8720C) and it is shown in Fig. S14c on a Smith chart. The reflection coefficient of the antenna indicates that it operates from 2.36 GHz to 2.61 GHz, which fully covers the ISM band used. We carried out similar

impedance measurements in other body phantoms (such as cadavers of mice and rats) and found that this matching network performed well in most subcutaneous implantation locations.

**Design of wireless powering unit.** To enable wireless powering of the implantable platform, we designed a wireless powering unit operating at the high-frequency RFID band of 13.56 MHz. The unit utilizes the inductive (or magnetic) coupling between two coils as the basic principle for wireless power transfer.

The unit consists of an external transmitter coil (or primary coil) connected to a high-frequency power source and a receiver coil (or secondary coil) on the implant[37,38]. The transmitter coil is made of copper wire with three turns and has a diameter of 10 cm. Two designs of the receiver coil were used in i-WPS and miniaturized i-WPS, respectively. Both coils have a size of 22 mm × 14 mm, but differ in the number of turns, metal coverlay, line width, and gap distance (see Figs. 5f and 6a for details). The structural design and optimization of the coils were carried out by full-wave simulations using CST Studio Suite. As illustrated in Fig. S17a, b, the surface current distribution on both coils is uniform in both straight lines and bends, at 13.56 MHz. In order to tune the coil to resonate at 13.56 MHz, we simulated the input impedance of the coil in vacuum (see Fig. S17c, d), which provided a reference value for the shunt capacitor at the terminal of the coil. In practice, the resonant frequency of the coil is related to both the implantation location and the encapsulation thickness, due to the variation of the inter-turn parasitic capacitance. As a result, in the realization of miniaturized i-WPS, a 56-pF capacitor is adopted.

**Fabrication and assembly of flexible i-WPS.** The fabrication process of the flexible test sensors is illustrated in Fig. S11. The flexible platform was fabricated as follows. On a 4-inch Si wafer, 10-μm thick polyimide (HD Microsystems GmbH, catalog no. PI2611) was spin-coated, soft-baked, and cured. A metallization of Ti (20 nm)/Au (5 μm) was performed by reactive sputtering (Alliance Concept AC450) and the patterning was carried out through liftoff, using a thick resist (MicroChemicals, catalog no. AZ10XT). A second layer of polyimide (5 μm) was spin-coated and cured on top of the metallization, and patterned through another photolithography (MicroChemicals, catalog no. AZ10XT) and RIE-etching (Corial 210IL) to expose all the connection points. After, laser-cutting (Optec WS Turret 200), the devices were peeled off. The disposable Mg inserts were fabricated in the same way as the Mg test sensors, but the design was for the space available in the flexible wireless platform. Before assembling the system, the selected encapsulations were deposited on the Mg inserts, protecting the connection pads.

For the assembly, the flexible platform was positioned and fixed with tape on a glass slide with a PI sheet in between to avoid the attachment of the system to the glass during soldering. Sn/Bi/Ag solder paste (SMDLTLFP10TS, Chipquik) bumps were dispensed manually on the connection pads exposed, and afterward, all the electrical components were positioned using pick-and-place equipment (JFP Microtechnic). A reflow at 200 °C of the solder paste ensured the soldering of all the components. The solder paste was also dispensed on the pads of the Mg inserts which were then flipped and aligned with the pads present on the flexible platform. Thus, air-gun soldering was performed. To guarantee mechanical fixation and protection for all the components of the system, the flexible platform was covered by a layer of silicone and a second layer of poly(isobutylene) (PIB, Oppanol, BASF) dissolved in cyclohexane (Sigma-Aldrich) at 20%wt.

**SEM, FIB milling, and EDX analysis.** SEM images of all the samples were acquired with a Zeiss GeminiSEM 300 microscope using an annular detector at a beam energy of 3 keV. A 4-nm thick gold layer was preliminarily deposited with a Quorum Q300T Sputter Coater to avoid charging effects during SEM observation.

The cross sections of the flexible Mg test vehicles were obtained by FIB milling in a dual beam equipment (Zeiss CrossBeam 540) equipped with Oxford Inst. EDX detector. The cross-section was milled with a current of 7 nA using a trapezoidal window and a second polishing step was performed at 1.5 nA using a narrow rectangular window: the milling depth was always in the range of 8–10 μm and the milling width was 20 μm. EDX analysis was performed (with 2 pA current and energy range of 20 keV) on the FIB-milled cross sections to obtain a compositional mapping of the chemical elements present.

**AFM.** AFM images of the Mg morphology before, during, and after corrosion were acquired with a Bruker FastScan AFM microscope in contact and tapping mode.

**Deposition of standard and hybrid encapsulations.** The standard barrier encapsulations selected for this work (i.e., PI, PDMS, and parylene C) were deposited through the usual cleanroom processes. PI was part of the fabrication of the Mg test sensors: it was spin-coated, soft-baked at 70 °C for 3 min and at 110 °C for 3 min, then hard-baked in a nitrogen environment. PDMS was first prepared by mixing a prepolymer precursor and a curing agent (ratio 10:1), and then it was spin-coated and cured at 95 °C for 1 h. Before the spin-coating, the pads of the Mg test sensors were covered with tape and this was peeled off before curing. Parylene C was deposited through RT-CVD following the Gorham route in a Comelec C-30-S machine.

The hybrid multilayer encapsulations were deposited in a single chamber, from the Comelec C30H equipment platform. The deposition of the organic layers (Parylene C) occurred at 30 °C whereas that for the ALD inorganic layers ($Al_2O_3$ and $TiO_2$) was set with a limit temperature of 100 °C in order to minimize thermal annealing effects onto the organic films. Nitrogen was used as purge gas together with all the gas precursors. The growth per cycle for $Al_2O_3$ and $TiO_2$ were 1.7 Å and 0.76 Å, respectively. To provide a conformal deposition of the encapsulations on the devices fabricated in this work, they were hung vertically in the chamber.

**Mechanical buckling tests.** The Mg devices were clamped on a homemade uniaxial stretcher and stretched at 10% of their initial length at 1 Hz (1 stretching cycle per second). The electrical resistance was measured using a 4-point probe configuration and a source meter (Keithley 2400). The stretcher and sourcemeter were both controlled using a custom-made GUI interface based on LabView 2015.

**In vitro experimental characterization.** In order to validate the designed system and evaluate the long-term functionality of the fabricated devices (Mg test sensors), accelerated aging experiments were carried out. Soaking tests in PBS with a pH of 7.4 (1×, Gibco), at different temperatures (37 °C, 67 °C, and 85 °C, respectively), were performed, using sealed glass containers inserted in a ventilated climatic chamber (ClimeEvent Weisstechnik; Heraeus HC7015) or on a hot plate.

**Real-time resistance and frequency monitoring.** The variation of the Mg electrical resistance was monitored over time. The resistance was monitored by measuring the voltage while applying a current of 1 mA. The interface was controlled and synchronized using a microcontroller and customized software based on LabVIEW 2015 (National Instruments), and it was able to provide a sampling rate of 0.03 Hz for the same Mg structure. In order to protect the contact points between the Mg structures and the connection tracks, a 15 wt% solution of poly(isobutylene) (PIB, Oppanol, BASF) dissolved in cyclohexane (Sigma-Aldrich) was drop cast onto those points via pneumatic printing: after curing at room temperature a thin layer of high-barrier sealant formed which delayed water permeation. Finally, the connections were embedded in epoxy to prevent any failure or damage during soaking.

The same procedures were followed for the frequency measurements which were performed simultaneously with the resistance monitoring.

**Ex vivo and in vivo-like experimental characterization.** Ex vivo measurements were performed on explanted chicken tissues which were previously submerged into a physiological solution and kept in a refrigerated environment. The measurements consist of positioning the tissue in a PBS bath kept at body temperature (approximately 37 °C). The PBS was aimed at keeping the tissue always hydrated. In order to avoid improper modifications of the mechanical and viscoelastic properties of the tissue and keep them close to the in vivo-like properties, before the measurement, it was removed from the refrigerated environment and kept at room temperature until thermal equilibrium. A hot plate was then used to control and monitor the temperature of the fluid from -25 °C to 37 °C at a rate of 1 °C/3 min. The characterization of the flexible platform followed the same equipment and methods as in vitro.

The in vivo-like measurements were performed on mice bodies (adult male or female C57BL/6 mice (body weight 18–35 g, age 8–25 weeks)): for this purpose, before the experiments, animals were deeply anesthetized and killed with intravenous (IV) injection of the ketamine-xylazine mixture (150 mg/kg and 10 mg/kg, respectively). The bodies were kept in a chamber with controlled temperature (body temperature, 37 °C) during the experiments. The characterization of the implanted flexible platform followed the same equipment and methods as in vitro. In order to make the high-frequency power distribution uniform, we also used a coupling coil tuned for the circuit's resonant frequency with a shunt capacitor. All the procedures adopted (i.e., housing, surgery, recordings, and euthanasia) were performed in compliance with the Swiss Veterinary Law guidelines and approved by the Veterinary Office of the Canton of Geneva (see procedures in previous works[39]).

**Reproducibility.** All data are collected as mean ± s.d. unless stated otherwise, and measurements were taken from distinct samples.

### Reporting summary
Further information on research design is available in the Nature Portfolio Reporting Summary linked to this article.

## Data availability
All data supporting the findings of this study are available within the paper and its Supplementary Information. Source data are provided with this paper.

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

## Acknowledgements
The authors would like to acknowledge financial support by InnoSuisse grants 45944.1 IP-ENG "FLEXCAN: flexible encapsulation of active implants", S.P.L. The authors would like to thank the staff at the Neural Microsystems Platform of the Wyss Center for Bio and Neuroengineering for their help with the fabrication processes; the staff at the Interdisciplinary Center for Electronic Microscopy (CIME) of EPFL for assistance in SEM observations; Ms. Marion von Allmen and Dr. Matthias van Gompel from Comelec SA for the deposition of the multilayer encapsulations; Mr. Ivan Furfaro from EPFL/LSBI and Mr. Cédric Meinen from EPFL/GR-KA, for providing assistance and tools to conduct electrical experiments.

## Author contributions
M.M. and J.D.R. conceptualized the experiments; S.P.L. and Y.L. supervised the work and acquired funding. F.C. and M.M. designed and fabricated samples for measurements and the implantable platform, and carried out in vitro and ex vivo testing. F.C. and J.D.R. designed, assembled, and calibrated the backscattering system. F.C. and M.M. conducted investigations, verification, and data curation. M.G. and A.S. provided the design of implantable antennas for wireless communication and coils for wireless power delivery. M.M. wrote the original draft. J.D.R. and M.G. contributed to the writing of the original draft. M.M. made the 3D-rendered illustrations. All authors contributed to reviewing and editing the manuscript.

## Competing interests
The authors declare no competing interests.
