## [Peer Review File · Nature Communications]

Wireless, battery-free, and real-time monitoring of water permeation across thin-film encapsulationREVIEWER COMMENTS

Reviewer #1 (Remarks to the Author):

This paper introduces the wireless system that can monitor the WTR of encapsulation through the hydrolysis of Mg electrode. In general, the manuscript provides a reasonable explanation of electrical characteristics as the sensing component. Nevertheless, there are areas in the manuscript that require further clarification and improvement, as outlined in the following comments.

* The WTR calculation in the manuscript is influenced by the encapsulation's defects, potentially leading to a misleading representation of the actual WTR value. It is essential to determine whether the Mg platform can detect any defects in the encapsulation. This is significant, considering that numerous encapsulations produced in academic facilities may have inherent defects.

* Figures 4a and 4b illustrate the alteration in oscillation frequency attributed to the variation in Mg resistance. However, the observed change appears quite minimal, and it's possible that this alteration could be influenced by rotation, bending, or tilting caused by animal movement during testing. Could the author please provide information regarding the stability of the measurement system to address this concern?

* It is difficult to envision how the system can be integrated with other implantable electronic systems. Is this system compatible solely with battery-powered systems, or can it also operate with other wireless, battery-free systems such as NFC-operated implantable electronics? This question is crucial as this concept could be valuable for determining the lifespan of implantable electronics, yet it may also potentially hinder the proper operation of other integrated implantable electronics.

* The actual demonstration appears to be conducted on the subcutaneous part of a relatively stationary animal for a specific period. However, many implantable electronics are typically implanted deep within the body. Could biofluid and other ions present deep in the body potentially affect wireless communication? It seems that the animal demonstration does not adequately explain the functionality and reliability of the system, particularly in scenarios where implantable devices are situated deeper within the body.

Reviewer #2 (Remarks to the Author):

In this manuscript, the authors present a wireless platform to monitor the lifetime of encapsulation in real time. The paper layout constructs quite clear and is well-written; however, it has some limitations, as described below, this reviewer recommends publication after addressing the following issues:

Major:

1. In bioelectronic devices, there are many ways of wireless signal transmission. Why did the authors select this transformation method? Although this communication method is wireless and battery-free, it requires an external RF source and RF receiver. In contrast, NFC communication and BLE communication based on wireless power supply only require a

mobile device (phone) to read the resistance value of magnesium. Therefore, you need to elaborate on the advantages and irreplaceable aspects of backscatter communication.

2. Why does a little decrease in amplitude between 10h and 12h appear in Figure 5f? Please explain it.

3. In the supplementary materials of the article, the schematic diagram of the fabrication of flexible Mg test sensors (Figure S2) should preferably include a description of what materials are represented by each color to make it easier for the reader to read and understand.

4. The format of the image titles in the supplementary materials should be consistent, such as Figure S6, S9, S13, S14, S15, and S16.

Minor:

Some spelling mistakes need to be corrected:

Line 59, "In this Article" should be written "In this article".

Line 408, "using an analytical model" should be written "Using an analytical model".

Response to Reviewer #1

This paper introduces the wireless system that can monitor the WTR of encapsulation through the hydrolysis of Mg electrode. In general, the manuscript provides a reasonable explanation of electrical characteristics as the sensing component. Nevertheless, there are areas in the manuscript that require further clarification and improvement, as outlined in the following comments.

We thank the Reviewer for their comments. We followed their suggestions adding clarifications, improvements and substantial modifications; thus, we hope that the amended manuscript will be now suitable for publication.

* The WTR calculation in the manuscript is influenced by the encapsulation's defects, potentially leading to a misleading representation of the actual WTR value. It is essential to determine whether the Mg platform can detect any defects in the encapsulation. This is significant, considering that numerous encapsulations produced in academic facilities may have inherent defects.

This point raised by the Reviewer is a very important aspect of the proposed technique based on Mg corrosion. We discussed in details the properties and applicability of the water-permeation sensing method based on Mg corrosion in one of our previous works (<https://doi.org/10.1002/adfm.202315420>). We proposed this method, called Magnesium test, as an alternative to all the existing water-permeability-measuring systems which are bulky, not sensitive enough, and not compatible with microfabrication and implantable devices. The Mg hydrolysis induced by water permeation through the thin-film encapsulation is analysed and the corrosion rate is correlated with the barrier performance through the analytical model. The method was validated through integration into real bioelectronic implants (i.e. electrocorticography devices and optoelectrodes) tested in different conditions and tissues. In the present manuscript the hydrolysis of Mg is initiated by the contact of water molecules on its surface, and which induce the localised oxidation of Mg and conversion into hydroxide.

The pathway travelled by the water molecules before touching the Mg surface depends on the type of encapsulation. In the case of (defect-free) polymers, water transport usually occurs predominantly by diffusion through the bulk. Exposure of the external surface of the polymer encapsulation to a constant relative humidity induces a uniform propagation front of water molecules towards the Mg surface, thus ideally inducing a homogeneous transformation of Mg into Mg(OH)₂ along the exposed Mg area. In the case of inorganic coatings that are intrinsically impermeable, water transport occurs instead through film defects such as pinholes or cracks. Such a defect-dominated mechanism of water transport induces a non-uniform propagation of water molecules, with preferential pathways/channels identified by the defects. This confers a non-homogeneous initiation of the corrosion process, which appears localised in correspondence to the defects and then spreading along the Mg surface. Hybrid organic-inorganic multilayer encapsulations exhibit both types of transport mechanisms. In fact, water molecules diffuse through the organic layers and permeate through the defects formed in the inorganic layers, with a resulting tortuous pathway.

To confirm these statements, we added a supplementary note (reported at the end of this response) where we show two Mg flexible sensors coated with polyimide and with a hybrid multilayer structure based on parylene C and an inorganic Al₂O₃ film. It can be observed (optical analysis) that with the polyimide encapsulation, the Mg surface is uniformly oxidized, while with the organic-inorganic encapsulation some spots of localised corrosion appear.

From the electrical point of view, the Mg corrosion induces an increase in the Mg resistance, irrespective of the type of encapsulation and the proposed wireless Mg platform can thus detect the water permeation in the encapsulation, leveraging the potential of the Magnesium test. The difference between the homogeneous diffusion through a polymer encapsulation and the inhomogeneous, defect-dominated water permeation through the organic-inorganic encapsulation lies in the shape of the resistance and WTR curves, as shown in Figure S10c,d. In fact, in the case of polyimide, the curve presents a smooth monotonic shape, with fluctuations only due to the sensitivity/electrical noise

in the used instrumentation, whereas in the case of the defective multilayer encapsulation the curve is very noisy, characterized by multiple “steps” or “peaks”, which correspond to the localised initiation of Mg corrosion.

Page 16, line 300

“It should be noticed that the proposed method is sensitive enough to discriminate between homogeneous diffusion through polymer encapsulations, and inhomogeneous diffusion through defective organic-inorganic multilayer encapsulations. In the latter case, inorganic layers contain inherent defects such as pinholes and microcracks that provide a pathway for rapid water permeation and localized Mg corrosion. These surface defects, especially pinholes, may also be present in polymer encapsulations as a result of non-ideal deposition processes. These two diffusion regimes are highlighted in the Supplementary Note 9 and in the Theory section.”

“Supplementary Note 9
 Defect-dominated water permeation

Figure S10. (a, b) Photos of Mg sensors soaked in PBS at 85°C, encapsulated with a 5 μ m-thick polyimide film and removed from PBS after 10h (a), or encapsulated with a $\sim 3\mu$ m-thick 3-dyad parylene C/Al₂O₃ multilayer on a polyimide substrate and removed from PBS after 500h (b). (c, d) Resistance curves for the two Mg sensors depicted in (a, b), respectively. The curve in (c) exhibits a smooth monotonic resistance increase due to diffusion-dominated corrosion; the curve in (d) exhibits a fragmented resistance increase due to defect-dominated corrosion.”

To summarize, the WTR platform can detect the density of defects in the encapsulation, that may result from the deposition process and environmental conditions. These defects, especially pinholes within inorganic layers, edge defects or defects at the organic/inorganic and Mg/encapsulation interfaces provide pathways for rapid water permeation and localised Mg corrosion. The influence of the defect population is therefore integrated into the temporal variation of the resistance of the Mg sensor from which the WTR is calculated. Therefore, the reported WTR values accurately reflect the barrier performance of a given encapsulation, and the detailed analysis of the corrosion process enables to characterize its defect population. To account for the defects in the encapsulation and quantify to which extent the tested encapsulations deviate from an ideal and defect-free barrier, we formulated an additional model that calculates first the overall resistance of the Mg film in the presence of a given density of defects, and then extracts the density of defects from the measured WTR. This model is described in the Theory section as follows:

Page 13 (SI), line 794:

“Model of Mg corrosion that accounts for defects in the encapsulation. *The previous model of Mg corrosion considers the encapsulation as defect-free with homogeneous water diffusion. However, in reality encapsulations can present a certain density of pristine defects that may result from the deposition process and environmental conditions. These defects, especially pinholes within inorganic layers, edge defects or defects at the organic/inorganic and Mg/encapsulation interfaces provide pathways for rapid water permeation and localised Mg corrosion (Erreur. L'origine riferimento non è stata trovata.(a)). The influence of the defect population is therefore integrated into the temporal variation of the resistance of the Mg sensor from which the WTR is calculated. This means that the WTR, acquired experimentally as proportional to the temporal variation of the Mg resistance, is averaged over the film as a whole, including the defects, and results in a higher value compared to the WTR of the defect-free encapsulation.*

Although there are many previous examples of models to account for the effect of local flaws in polymeric encapsulations [41,42], we propose here a simplified model that is adapted to the Mg electrical test.

Let us consider an encapsulation with a given fraction of defective area, Δ :

$$\Delta = \frac{S^D}{S} \quad (35)$$

where S^D, S are the total defects' surface and the total surface of the encapsulation. Below each defect in the encapsulation, there will be a corroded area (called hereafter “spot”) in the Mg film. We assume that the corroded spots spread through the thickness of the Mg film. As sketched in Figure S1(a) the Mg film area is divided into spot-free areas and spot-containing areas, each of width w and perpendicular to the axis of resistance measurement, with isolated, non-overlapping circular spots of radius r . Each spot-containing area (width w , length $2r$) is split by each spot in two regions, whose electrical resistances are connected in parallel (assuming that the spot is empty of Mg). Thus, on a first approximation the conductance of one spot-containing Mg area is given by:

$$G^D = \frac{(w - 2r)h}{2\rho r} \quad (36)$$

where ρ is the Mg electrical resistivity and h is the Mg thickness. The resistance of all the spot-containing Mg areas is:

$$R_T^D = \frac{2\rho r\Delta}{(w - 2r)h} \quad (37)$$

The resistance of all the spot-free Mg areas is:

$$R_T^{DF} = \frac{\rho(L - 2r\Delta)}{wh} \quad (38)$$

Thus, the total Mg resistance and conductance are:

$$R_{Mg} = R_T^D + R_T^{DF} = \frac{\rho}{h} \underbrace{\left(\frac{L}{h} + \frac{4r^2 \Delta}{w(w-2r)} \right)}_{\xi} = \frac{\rho \xi}{h} \quad (39)$$

$$G_{Mg} = \frac{h}{\rho \xi} \quad (40)$$

From the Mg corrosion equation, one can show that the WTR of the encapsulation is given by:

$$WTR = -2 \left(\frac{M_{H_2O}}{M_{Mg}} \right) \frac{\delta L w}{S} \rho \xi \frac{dG_{Mg}}{dt} \quad (41)$$

where δ is the Mg mass density; M_{H_2O} , M_{Mg} are the molecular weights of water and Mg, respectively. We can write:

$$WTR = -2 \underbrace{\left(\frac{M_{H_2O}}{M_{Mg}} \right) \left(\frac{S_{Mg}}{S} \right) \delta \rho \frac{L}{w}}_K \cdot \frac{dG_{Mg}}{dt} - 2 \underbrace{\left(\frac{M_{H_2O}}{M_{Mg}} \right) \delta \rho \frac{4r^2}{w(w-2r)} \frac{\Delta}{1-\Delta}}_{K^D} \cdot \frac{dG_{Mg}}{dt} \quad (42)$$

$$WTR = K \frac{dG_{Mg}}{dt} + K^D \frac{dG_{Mg}}{dt} = WTR_{DF} + WTR_D \quad (43)$$

where K, K^D are two constants that only depend on physical constants, specimen geometry and defect density, and WTR_{DF}, WTR_D are the water transmission rates for the defect-free encapsulation and for the defects, respectively. By comparing the (5) and (43), it can be deduced that the proposed experimental method provides the WTR of the defect-free encapsulation ($WTR_{DF} = K dG_{Mg}/dt$): to obtain the value of the WTR corresponding to the encapsulation with pre-existing defects, the term $WTR_D = K^D dG_{Mg}/dt$, should be added. The fraction of defective area Δ can be quantified by optical observation and image processing (before implantation), and also the defect size (r) can be obtained with standard microscopy techniques (e.g. SEM, AFM); thus, K^D can be calculated and the WTR for the defective encapsulation can be extracted.

As an example, considering a defect size of $\sim 100 \mu m^2$ ($r \approx 5 \mu m$) in polyimide films with defect density of 0.2 cm^{-2} [43] (Figure S1(b)), then $\Delta = 2 \times 10^{-7}$, and (with $w = 100 \mu m$) $K^D = 5.35 \times 10^{-14} \text{ g}\Omega\text{m}^{-2}$. The low value of this coefficient compared to K ($\sim 10^{-4} \text{ g}\Omega\text{m}^{-2}$) confirms that for polymers such as polyimide, the contribution of defects (WTR_D) is minimal compared to the diffusive mechanism (WTR_{DF}).

In the case of more defective encapsulations, e.g. polymers coated with inorganic layers or hybrid organic-inorganic multilayers, the size and density of defects can be remarkable and lead to a deviation from the defect-free WTR (WTR_{DF}). The fraction Δ can be higher, due to the presence of multiple inorganic layers, therefore, higher values of K^D can be achieved. Figure S1(c) illustrates the variation of K^D as a function of Δ and r (the latter in the range of 0-10 μm and in 0-1 μm), for different values of the width w of the Mg film area. As expected, K^D increases both with Δ and r , and decreases with w , reaching peaks of $\sim 3 \times 10^{-4}$, $\sim 5 \times 10^{-6}$, $\sim 4 \times 10^{-8}$ (with $\Delta = 10^{-3}$, $r = 10 \mu m$) for $w = 25 \mu m$, $100 \mu m$, $1000 \mu m$, respectively. The value of WTR can thus deviate more or less remarkably from WTR_{DF} , depending on the order of magnitude of K^D . Considering for instance, a defect density of $\sim 100 \text{ cm}^{-2}$ for an ALD Al_2O_3 layer [42], then $\Delta = 10^{-4}$ and $K^D \approx 5 \times 10^{-9} \text{ g}\Omega\text{m}^{-2}$ (with $w = 100 \mu m$, $r = 200 \text{ nm}$). This value is noticeably higher than that for a standard polymer, but it is still negligible if compared with the order of magnitude of WTR_{DF} .

The density of defects can also be extracted by measuring the apparent WTR of the unencapsulated Mg: this transmission rate is the same as WTR_D , since in correspondence to the defects of the encapsulation, Mg is not coated.

Figure S1. (a) Scheme of defects present in the thin-film encapsulation coating the Mg film. (b) AFM topography image showing three corrosion spots in the Mg film due to the presence of pinhole defects in the encapsulation (removed before the imaging). (c) 3D plots of K_D (units $\text{g}\Omega\text{m}^{-2}$) as function of the defect radius and the defect density (fraction Δ), for different widths of Mg tracks (25, 100, 1000 μm).

* Figures 4a and 4b illustrate the alteration in oscillation frequency attributed to the variation in Mg resistance. However, the observed change appears quite minimal, and it's possible that this alteration could be influenced by rotation, bending, or tilting caused by animal movement during testing. Could the author please provide information regarding the stability of the measurement system to address this concern?

We appreciate that this aspect was not sufficiently clarified in the manuscript. From the perspective of theory of wireless antennas, rotation or tilting may only change the radiation pattern of the implanted device and bending might also slightly change the tuning of the antenna, but neither of them can change the operating frequency of the transmitted signal. Since the implanted antenna used in the wireless system has sufficient bandwidth, it can be stably used for wireless telemetry of changes in oscillation frequency.

The supplementary movies help showing that the RF signal is the same during tilting, rotation, in-plane translations. Therefore, minimal variations of the RF oscillation frequency could be attributed to bending of the Mg resistor, which causes an alteration of its resistance of at most ~0.4%. The corresponding change in the oscillator frequency can then be predicted by equations [4-5]. In fact, it may be written:

$$f_{osc} = \frac{\psi}{R_{set}} \Rightarrow f'_{osc} = \frac{\psi}{R'_{set}}$$

where

$$R'_{set} = \frac{R_{fix}(R_{fix} + R_{Mg} + \Delta R)}{2R_{fix} + R_{Mg} + \Delta R}$$

with ΔR the change in the Mg resistance due to bending, and R'_{set}, f'_{osc} are the set resistance and oscillation frequency for the bent Mg resistor.

In the manuscript, we explain that (page 9, line 183): *“a shift in f_{osc} can be achieved of ~35 Hz for a resistance change of 1 Ω ”*. We made a set of measurements for bending the Mg resistance at different bending ratios (or radii of curvature), finding that it can change of at most 0.5 Ω . The corresponding variation of ~35 Hz is thus not enough to be detected by the system, considering that the measured value of the oscillating frequency was in the order of GHz or MHz. Therefore, the observed change, reported in figures 4a and 4b can only be attributed to the variation of Mg resistance due to corrosion. To confirm this, we provide two sets of measurements of the same wireless device but with different bending ratios of the Mg resistor: it turns out that the resulting signal and its variation in time are very similar.

We clarify these details in the amended manuscript, adding a supplementary note and an explanation in the main text.

Page 13, line 234

“Slight variations in the signal amplitude or in the oscillation frequency might be ascribed to bending of the Mg sensors (see the Supplementary note 5).”

“Supplementary note 5

Figure S6. (a) Mechanical setup for cyclic buckling of Mg sensor. On the right the scheme shows the bending radius exhibited by the Mg sensor. (b) Cyclic buckling test (15 mm displacement, 1 Hz frequency) and simultaneous monitoring of Mg electrical resistance. (c) Dependence of Mg resistance on bending radius: the resistance increases within 0.1Ω . (d) Backscattered signals for a Mg sensor with two bending radii, i.e. 0mm and 2.04 mm. There is no detectable difference between the two signals: although the different Mg resistance induces a shift in the oscillation frequency, this shift is not detectable because very low compared to the value of the oscillation frequency.

Page 21, line 408

“In the Supplementary Note 18, the contribution of bending, rotation or tilting caused by any animal movement during testing is discussed.”

“Supplementary note 18

Contribution of bending, tilting, rotation of the animal body.

Figure S19. Backscattered signals detected by the mini-i-WPS implanted in the animal body subjected to different movements: in-plane translation, out-of-plane translation, rotation, bending. Although minimal variation in the signal amplitude can be observed, there is no variation in the oscillation frequency, providing a robust validation for the proposed wireless platform.

* It is difficult to envision how the system can be integrated with other implantable electronic systems. Is this system compatible solely with battery-powered systems, or can it also operate with other wireless, battery-free systems such as NFC-operated implantable electronics? This question is crucial as this concept could be valuable for determining the lifespan of implantable electronics, yet it may also potentially hinder the proper operation of other integrated implantable electronics.

This is a very good point raised by the Reviewer. The wireless platform proposed in this work is meant for the accurate detection of water permeation through thin-film encapsulations of implantable bioelectronics. Therefore, it has been considered as a single complete implantable device and the manuscript is focused on describing this specific functionality. However, the size of the design is really driven by the need to use discrete components for prototyping. While we adopted specific frequencies for wireless power transfer and wireless data communication (i.e. 13.56 MHz and 2.4 GHz respectively), the proposed concept is expected to work for higher frequencies, thus enabling further antenna miniaturization (e.g., 5.8 GHz). Additionally, the components of the overall electronics are small and simple, lending themselves to easy miniaturization in an application specific integrated circuit (ASIC). One possible embodiment would be to create an ASIC of the design as a small sticker that could be independently added onto an implant with its own functionality, such as an RFID tag is integrated into clothing tags. In that case, a single frequency for wireless powering and backscattering communication may be used to simplify the operation. Furthermore, the Mg resistor, that in the current design is shaped as an external insert with macroscopic dimensions, can be miniaturized and included in the overall circuitry on a microscopic scale (e.g. as a small serpentine-like resistor with $\sim 1\text{-}10\ \mu\text{m}$ trackwidth). In this way, a full bioelectronic implant would include this compact module to monitor the health of the encapsulation without interfering with the rest of the functional system.

Thanks to developments creating Bluetooth Low Energy (BLE)-compatible backscatter systems (10.1109/WISNET.2019.8711794, <https://doi.org/10.1109/RFID54732.2022.9795961>, 10.1109/TMTT.2019.2938162), a design could be envisioned that could be interrogated and read with smartphones.

We included this explanation in the amended version of the manuscript, to clarify that the proposed system can effectively be integrated with other wireless, battery-free systems in more complex implant designs.

Page 25, line 489

“Although this work is focused on the characterization of the WPS platform as a single implant, it can be easily integrated into more complex wireless systems with other functionalities. The size of the design, in fact, has been driven by the need to use discrete components for prototyping. While we adopted specific frequencies for wireless power transfer and wireless data communication (i.e. 13.56 MHz and 2.4 GHz respectively), the proposed concept is expected to work for higher frequencies, thus enabling further antenna miniaturization (e.g., 5.8 GHz). Additionally, the components of the overall electronics are small and simple, lending themselves to easy miniaturization in an application specific integrated circuit (ASIC). One possible embodiment would be to create an ASIC of the design as a small sticker that could be independently added onto an implant, such as an RFID tag is integrated into clothing tags. Furthermore, the Mg resistor, that in the current design is shaped as an external insert with macroscopic dimensions, can be miniaturized and included in the overall circuitry on a microscopic scale (e.g. as a small serpentine-like resistor with $\sim 1\text{-}10\ \mu\text{m}$ trackwidth). In this way, a full bioelectronic implant would include this compact module to monitor the health of the encapsulation without interfering with the rest of the functional system. Thanks to developments creating BLE-compatible backscatter systems [17,32,33], an optimized design could be envisioned that could be interrogated and read with smartphones.”

* The actual demonstration appears to be conducted on the subcutaneous part of a relatively stationary animal for a specific period. However, many implantable electronics are typically implanted deep within the body. Could biofluid and other ions present deep in the body potentially affect wireless communication? It seems that the animal demonstration does not adequately explain the functionality and reliability of the system, particularly in scenarios where implantable devices are situated deeper within the body.

We acknowledge that deeply implanted bioelectronics may experience significant attenuation of wireless link efficiency due to lossy biological tissues, particularly in-body path loss, leading to very tight link budgets. According to previous theoretical studies on implantable antennas (<https://doi.org/10.1109/OJAP.2023.3276686>; <https://doi.org/10.1109/TMTT.2022.3231492>), especially for deep-body implants, the in-body path loss can be analytically decomposed into three main contributions: (i) reactive near-field losses, (ii) propagating field loss and (iii) reflection loss occurring at the tissue interface. The first two of them, in particular, will cause more losses as the implantation depth increases. As a result, the gain of the antenna decreases significantly from subcutaneous implantation to deep implantation, which poses a challenge to the link strength of wireless communications. However, this is only a change in signal strength and does not affect the oscillation frequency of backscatter communication. Therefore, we believe that this scenario for a subcutaneous implantation is sufficient.

In addition, for the engineering design of implantable RF devices, it is often necessary to optimize the antenna structure and matching network according to the specific implantation location and packaging structure to reduce mismatch losses of the implantable antenna. Therefore, if the implant is designed for deep implantation, the impact of unnecessary reflection losses on wireless communications can be avoided by redesigning and retuning the antenna used in the implant, based on accurate 3D electromagnetic simulations of the host tissue.

Finally, biofluids and other ions present deep in the body can permeate through the thin-film encapsulation and the proposed wireless system is aimed at evaluating this water/ions ingress and the wireless operation remains intact before the failure of the encapsulation.

We clarified this point in details in the amended version of the manuscript.

Page 21, line 410

“Moreover, the in vivo-like demonstration was conducted for subcutaneous implantation, but deeply implanted bioelectronics may experience significant attenuation of wireless link efficiency due to lossy biological tissues, particularly in-body path loss. According to previous theoretical studies on implantable antennas [28,29], especially for deep-body implants, the in-body path loss can be analytically decomposed into three main contributions: (i) reactive near-field losses, (ii) propagating field loss and (iii) reflection loss occurring at the tissue interface. The first two of them, in particular, will cause more losses as the implantation depth increases. As a result, the gain of the antenna decreases significantly from subcutaneous implantation to deep implantation, which poses a challenge to the link strength of wireless communications. However, this is only a change in signal strength and does not affect the oscillation frequency of backscatter communication. In addition, for the engineering design of implantable RF devices, it is often necessary to optimize the antenna structure and matching network according to the specific implantation location and packaging structure to reduce mismatch losses of the implantable antenna. Therefore, if the implant is designed for deep implantation, the impact of unnecessary reflection losses on wireless communications can be avoided by redesigning and retuning the antenna used in the implant, based on accurate 3D electromagnetic simulations of the host tissue.”

Response to Reviewer #2

In this manuscript, the authors present a wireless platform to monitor the lifetime of encapsulation in real time. The paper layout constructs quite clear and is well-written; however, it has some limitations, as described below, this reviewer recommends publication after addressing the following issues:

We are grateful for the Reviewer's comments and suggestions. We revised the manuscript so we hope that it will be suitable for being published.

Major:

1. In bioelectronic devices, there are many ways of wireless signal transmission. Why did the authors select this transformation method? Although this communication method is wireless and battery-free, it requires an external RF source and RF receiver. In contrast, NFC communication and BLE communication based on wireless power supply only require a mobile device (phone) to read the resistance value of magnesium. Therefore, you need to elaborate on the advantages and irreplaceable aspects of backscatter communication.

We thank the Reviewer for this valuable comment and suggestion. Backscatter communication (BC) has several unique and irreplaceable advantages compared to other wireless systems like NFC and BLE, particularly in the context of implantable devices. These pros particularly regard power efficiency, simplicity and long-term viability, which make BC an invaluable technology for medical implants, providing a sustainable and safe communication method that is difficult to match with NFC or BLE.

In terms of power efficiency, BC adopts RF signals, reflecting them to communicate data, significantly reducing the power requirements since the device does not generate its own RF signal, eliminating the need for an internal power source. BC also can leverage existing RF infrastructure (like WiFi, cellular, or dedicated RF sources), making it adaptable to various environments without the need for specific readers or transmitters, used by NFC and BLE. Since BC devices don't need complex RF transmitters, the overall design is simpler, leading to less complex implantable devices. Additionally, the minimal power usage of BC results in lower heat generation. Excess heat can be harmful to body tissues, so the low-heat characteristic of BC makes it safer for long-term implants. Moreover, BC systems typically generate less electromagnetic interference, reducing potential risks to other sensitive electronic medical devices and reducing potential adverse effects on human tissues. As a further note, although BLE and NFC have their specific range benefits, BC can operate effectively over longer distances without the need for precise alignment (unlike NFC which requires very close proximity, even with magnetic resonance coupling), and this can be beneficial in real-world medical scenarios.

To summarize, for this work we chose to use BC due to its significant energy efficiency relative to conventional RF solutions, such as BLE, and its potential for far-field wireless communication as would be needed for in vivo experiments with animals in their home cages. BLE has been demonstrated in many small implants and could be an option for a future embodiment of the device. For this work, as a proof-of-concept demonstration, analogue BC of a subcarrier frequency provided an appealing balance between power consumption, measurement resolution, cost, and complexity. NFC itself uses BC, however, it does so in the 13.56 MHz frequency band with near-field electromagnetic waves requiring very close proximity for operation. As a result, using NFC communication in future applications with moving rodents could be problematic.

To clarify this point, we added the following explanation in the amended manuscript.

Page 24, line 475

"In this context, the choice of using backscatter communication has been driven by its significant energy efficiency relative to conventional RF solutions, such as Bluetooth-Low-Energy (BLE), and its potential for far-field wireless communication as would be needed for in vivo experiments with animals in their home cages. BLE has been demonstrated in many small implants [31] and could be an option for a future embodiment of the device. For this work, as a proof-of-concept demonstration, analogue backscattering of a subcarrier frequency provided an appealing balance between power consumption, measurement resolution, cost, and complexity. Near-field communication (NFC), could be considered as an

alternative strategy for wireless operation and it also uses backscattering; however, it does so in the 13.56 MHz frequency band with near-field electromagnetic waves requiring very close proximity for effective operation. As a result, using NFC communication in future applications with moving rodents could be problematic.”

2. Why does a little decrease in amplitude between 10h and 12h appear in Figure 5f? Please explain it.

A decrease in the signal amplitude might be caused by small changes to the measurement environment, e.g. slight change in the antenna orientation or dispersion in the dielectric properties of biological tissues caused by temperature changes, both of which may cause fluctuations in the path loss of the considered wireless link. As long as the wireless link budget remains sufficient, this will not have an impact on backscatter communication. This highlights the benefit of encoding information in the frequency domain rather than in amplitude, as it provides greater resilience to these changes (analogously, we can think of the difference in audio quality between AM and FM radio stations).

We clarified this point in the amended manuscript.

Page 18, line 353

“Minimal alterations in the signal amplitude may be caused by small changes to the measurement environment, e.g. slight change in the antenna orientation or dispersion in the dielectric properties of biological tissues caused by temperature changes, which may induce fluctuations in the path loss of the considered wireless link. However, this does not have impact on the backscatter communication and highlights the benefit of encoding information in the frequency domain rather than in amplitude.”

3. In the supplementary materials of the article, the schematic diagram of the fabrication of flexible Mg test sensors (Figure S2) should preferably include a description of what materials are represented by each color to make it easier for the reader to read and understand.

We are grateful for this suggestion: we revised the figure accordingly, adding indication by arrows and a color legend for the materials involved in the fabrication process.

Old version (Figure S2):

New version (Figure S2):

4. The format of the image titles in the supplementary materials should be consistent, such as Figure S6, S9, S13, S14, S15, and S16.

We revised the format of the figure captions, for Figure S6, S9, S13, S14, S15 and S16: they are highlighted in yellow in the amended manuscript.

Minor:

Some spelling mistakes need to be corrected:

Line 59, "In this Article" should be written "In this article".

Line 408, "using an analytical model" should be written "Using an analytical model".

We revised the spelling typos, as follows.

Page 3, line 59: "In this article"

Page 23, line 439: "Using an analytical model"

REVIEWERS' COMMENTS

Reviewer #1 (Remarks to the Author):

Author addressed the previous comment accordingly.

Reviewer #2 (Remarks to the Author):

The authors have addressed all my questions in details. And I recommend this manuscript to be accepted as it is in this version.